A new neornithischian dinosaur from the Upper Jurassic Tiaojishan Formation of northern China

Yang Yunfeng 1 2
King James L. 2 3 4
Xu Xing xu.xing@ivpp.ac.cn 2 5
1 College of Earth and Planetary Sciences, University of Chinese Academy of Sciences , Beijing , China
2 Key Laboratory of Vertebrate Evolution and Human Origins, Institute of Vertebrate Paleontology and Paleoanthropology, Chinese Academy of Sciences , Beijing , China
3 Colorado Northwestern Community College , Craig , CO , United States of America
4 School of Earth Sciences, University of Bristol , Bristol , United Kingdom
5 Centre for Vertebrate Evolutionary Biology, Yunnan University , Kunming , China
Knoll Fabien
Electronic publication date: 2025 Jul 11
Publication date: 2025
Volume: 13
Electronic Location ID: e19664
Received 2025 Jan 29; Accepted 2025 Jun 5
Copyright: ©2025 Yang et al.
Copyright year: 2025
Copyright holder: Yang et al.
License: This is an open access article distributed under the terms of the Creative Commons Attribution License, which permits unrestricted use, distribution, reproduction and adaptation in any medium and for any purpose provided that it is properly attributed. For attribution, the original author(s), title, publication source (PeerJ) and either DOI or URL of the article must be cited.
License URL: https://creativecommons.org/licenses/by/4.0/

Keywords: Yanliao biota, Neornithischia, Laryngeal elements

Funding: National Science Foundation of China No. 42288201 The study was funded by National Science Foundation of China (No. 42288201). The funders had no role in study design, data collection and analysis, decision to publish, or preparation of the manuscript.

==============================
The Middle and Late Jurassic Yanliao Biota is different from other contemporaneous fossil assemblages in that it lacks neornithischian dinosaurs. Here, we report a new, early-diverging neornithischian, Pulaosaurus qinglong gen. et sp. nov., from the Upper Jurassic Tiaojishan Formation of Qinglong, Hebei Province, of northern China. Diagnostic or noteworthy morphological characteristics of P. qinglong include: five premaxillary teeth; a small boss is present on the caudoventral corner of the dorsal ramus of the jugal; a nuchal crest is located along the parietal; the manus has five digits; a supra-acetabular crest is present on the ilium; the paired arytenoids are gracile and leaf-like in form; the obturator process along the ischium is located near the pubic peduncle; a notch-like shaped obturator opening is present within the pubis; a robust fibular condyle forms a dorsoventrally extending crest on the tibia; a subtriangular flange on the anterior surface of the astragalus extends dorsolaterally along three distal tarsals; three of the distal tarsals are unfused, including a small drop-shaped distal tarsal 3; distal tarsal 3 is pierced by a foramen. A phylogenetic analysis places P. qinglong as one of the earliest-diverging neornithischians yet described. Moreover, P. qinglong represents the second known dinosaur to preserve ossified laryngeal elements, thus suggesting that a bird-like vocalization evolved early in non-avian dinosaur evolution.

Introduction

The Middle-to-Late Jurassic-aged Yanliao Biota is one of the most significant Mesozoic, terrestrial lagerstätte in China, with an age that ranges from 168 Ma to 157 Ma (Bai et al., 2024; Huang, 2015; Liu, Wu & Han, 2022; Zhou & Wang, 2017) and is comprised of fossil assemblages from the Jiulongshan and the Tiaojishan Formations (Boyd, 2015; Huang, 2015; Zhou & Wang, 2017). The Daohugou Biota, whose age was about 168-164 Ma, represents the early stage of the Yanliao Biota while the Linglongta Biota whose age was about 162-157 Ma represents its late stage (Huang, 2015; Zhou & Wang, 2017). In total, there have been 54 genera and 58 species of vertebrates reported from the Yanliao Biota, including nine species of non-avian dinosaurs (Liu, Wu & Han, 2022; Zhou & Wang, 2017). Because the Yanliao Biota preserves large amounts of vertebrate material from many taxa, it offers insight to major palaeobiological events, such as the temporal origin of birds and the early evolution of mammals (Liu, Wu & Han, 2022; Xu et al., 2022; Zhou & Wang, 2017). However, non-avian dinosaurs found in the Yanliao Biota are all small-bodied theropods whereas Ornithischia is represented by only one species, which may possibly be from the Jehol Biota (Zhou & Wang, 2017). This is in stark contrast to other contemporaneous Chinese terrestrial faunas such as the Shishugou and Shaximiao faunas where body size and taxonomic composition are more variable (Liu, Wu & Han, 2022; Xu et al., 2022).

Neornithischia is a significant group of dinosaurs whose earliest origin may be dated back to the Middle Jurassic, represented by several early-diverging taxa including Sanxiasaurus, Agilisaurus, Hexinlusaurus found in China (Fonseca et al., 2024; Li et al., 2019; Barrett, Butler & Knoll, 2005). Besides China, there have been reports of neornithischian fossils from the Middle Jurassic of Russia (Cincotta et al., 2019), Scotland (Panciroli et al., 2025) and strata from other geological times and countries. Neornithischia diverges rapidly into a number of taxa in Cretaceous (Fonseca et al., 2024). The phylogeny of early-diverging neornithischians which used to be assigned to ‘Hypsilophodontidae’ has been controversial in recent years, especially for the early-diverging taxa referred as (Boyd, 2015; Han et al., 2018; Dieudonné et al., 2020; Brown et al., 2022). There are two major hypotheses for the phylogeny of these early-branching neornithischians (Galton & Upchurch, 2004): (1) most of the early-diverging neornithischians are grouped within Cerapoda, as a paraphyletic group of early-diverging ornithopods (Brown et al., 2022); (2) most of the early-diverging neornithischians are outside Cerapoda and within the clade Thescelosauridae (Brown et al., 2022). Boyd (2015) and Han et al. (2018) agree on the former hypothesis while Dieudonné et al. (2020) and Fonseca et al. (2024) agree on the latter hypothesis; however, the controversy over the phylogeny of early-diverging neornithischians stems from the differences in the morphological characters selected in the different phylogenetic analyses (Brown et al., 2022), so the discovery of a new early-diverging neornithischian species helps to clarify relationships among early-diverging neornithischian taxa.

Here, we describe a new specimen found in the Upper Jurassic Tiaojishan Formation, County Qinglong, Province Hebei, China. This well-preserved specimen IVPP V30936 has both basal and derived characteristics of Ornithischia and preserves cololites, hyoids, and an ossified larynx. This new neornithischian specimen helps us understand the biodiversity of Yanliao Biota and the phylogeny of early-diverging Neornithischia better.

Materials & Methods

The specimen V30936 was collected by Mr. Yong Wang and later transferred to the Institute of Vertebrate Paleontology and Paleoanthropology (IVPP). It is currently stored in IVPP. It was found in southern Shimen Gou, County Qinglong, Province Hebei, People’s Republic of China. The specimen is fully prepared and based on close examinations and comparations of the osteology and matrix, the specimen was deemed authentic.

The anatomical information of IVPP V30936 was acquired by personal observations. They were supplemented by computed tomography (CT) scans of the pelvic girdle. The CT scans were conducted by 160-Micro-CL (computed laminography) in IVPP, which has generated 492 images. The maximal voltage for CT scans is 160 kv and the maximal current is 1.0 mA. The voxel size is 10 µm. No filter is used. The CT data provides insight into areas of the pelvic girdle obscured by the femora and the chevrons to gain more complete anatomical data on IVPP 30936.

The electronic version of this article in Portable Document Format (PDF) will represent a published work according to the International Commission on Zoological Nomenclature (ICZN), and hence the new names contained in the electronic version are effectively published under that Code from the electronic edition alone. This published work and the nomenclatural acts it contains have been registered in ZooBank, the online registration system for the ICZN. The ZooBank LSIDs (Life Science Identifiers) can be resolved and the associated information viewed through any standard web browser by appending the LSID to the prefix http://zoobank.org/. The LSID for this publication is: urn:lsid:zoobank.org:pub:D3939AEC-9C5B-4397-9BA4-47CB3F9DFEC8. The online version of this work is archived and available from the following digital repositories: PeerJ, PubMed Central SCIE and CLOCKSS.

Phylogenetic analysis

To assess the phylogenetic position of IVPP V30936 within ornithischians, we ran the phylogenetic analysis on the emended matrix dataset of Han et al. (2018), including IVPP V30936 and Sanxiasaurus modaoxiensis, a recently reported early-diverging neornithischian taxon. The coding of Sanxiasaurus was based on the description and images provided by Li et al. (2019). For consistency, we followed the phylogenetic nomenclature of higher-level ornithischian taxa as defined by Madzia et al. (2021).

The final dataset consists of 380 characters scored for 70 ingroup taxa and four outgroup taxa. The analysis was conducted in TNT with each character equally weighted. The analysis is based on a traditional search with 1,000 random seeds and 1,000 replications. The swapping algorithm is tree bisection reconnection (TBR), with 100 trees to save per replication. The maximal memory of trees was set to 100,000 and zero-length branches collapsed.

Unstable taxa were identified by the command ‘pruned trees’ and include four taxa, such as Pisanosaurus, Micropachycephalosaurus, Zephylosaurus, and Yueosaurus. Having excluded the unstable taxa and repeating the steps described above, the reduced strict consensus tree was gained and phylogenetic relationships of the remaining taxa were display.

Additionally, we carry out another two analyses in TNT based on the emended dataset of Fonseca et al. (2024) which consists of 943 characters and 173 taxa. In one analysis, each character is equally weighted while the weight with default function is set to be 12.0. The analysis is based on a traditional search with 1,000 random seeds and 1,000 replications. The swapping algorithm is tree bisection reconnection (TBR), with 100 trees to save per replication. The maximal memory of trees was set to 100,000 and zero-length branches collapsed.

Results

Systematics paleontology

Dinosauria Owen, 1842	
Ornithischia Seeley, 1887	
Neornithischia Cooper, 1985	
Pulaosaurus qinglong gen. et sp. nov.	

Etymology

The generic name is derived from Chinese Pinyin for “Pulao”, a mythical creature resembling the Chinese dragon. According to Chinese legends, the “Pulao” engages in loud shouting, thus referencing the possible bird-like vocalizations of this species. The specific name is derived from the Chinese Pinyin for “Qinglong”, which is the name of the county in Province Hebei, China, where the specimen was found.

Holotype

IVPP V30936 is comprised of a nearly complete skeleton prepared on a brownish-red sandstone slab. The specimen preserves most of the skull and a complete postcranial skeleton. Many skull elements are displaced and disarticulated due to mediolateral crushing. All cervical vertebrae and dorsal vertebrae are compressed and obscured. The sacrum is overlapped by the ilium. Only two proximal, five middle and 17 distal caudal vertebrae are well-preserved. Both scapulae and the sternum are broken with only fragments being preserved. The left forelimb is complete and articulated; however, the right humerus is not preserved and elements of the right manus are disarticulated. Elements of the pelvic girdle are preserved although many morphological characters are unknown due to overlapping and displacement of the available elements. Most of the hindlimbs are preserved except for the right distal tarsals 2, 3, 4; left calcaneum; and phalange 1 of the left pes. Additionally, a pair of ceratobranchials and ossified arytenoids are also preserved ventrally adjacent to the mandible. Cololites (gut contents) are preserved in the abdominal cavity.

Locality and horizon

Southern Shimen Gou, County Qinglong, Province Hebei, People’s Republic of China. Tiaojishan Formation, Callovian-Oxfordian (Middle-Upper Jurassic).

Diagnosis

A small-bodied neornthischian dinosaur characterized by the combination of the following characteristics (autapomorphies preceded by an asterisk): five premaxillary teeth; a small boss is located on the caudoventral corner of the dorsal ramus of the jugal; a nuchal crest that is located on the parietal the mani have five digits; a supra-acetabular crest is located on the ilium; a pair of gracile, leaf-like arytenoids are present; the obturator process is near the pubic peduncle; the opening of the obturator on the pubis is notch-shaped; *a robust fibular condyle forms a dorsoventrally extending crest on the tibia; a subtriangular flange extends dorsolaterally on four distal tarsals; *three distal tarsals are unfused with distal tarsal three that is drop-shaped; distal tarsal three is pierced by a foramen.

Because our phylogenetic analysis recovered Pulaosaurus qinglong in a systematic position near that of the contemporaneous Agilisaurus louderbacki (Peng, 1992) and Hexinlusaurus multidens (Barrett, Butler & Knoll, 2005), from Sichuan Province, southwestern China, we list additional taxonomical differences between Pulaosaurus and Agilisaurus and Hexinlusaurus. Pulaosaurus qinglong differs from Agilisaurus louderbacki and Hexinlusaurus multidens in the following characteristics: the relative length of the Pulaosaurus skull is more elongated while the skull of Agilisaurus is foreshortened; all of the premaxillary teeth in Pulaosaurus are subequal to each other in size while the middle premaxillary teeth are largest in Agilisaurus; a small boss is present on the jugal of Pulaosaurus while there is no ornamentation on the jugal of Agilisaurus or Hexinlusaurus; the ventral edge of the jugal is straight in Pulaosaurus but caudoventrally deflects in Agilisaurus and Hexinlusaurus; the orbital margin of the postorbital projects into the orbit in Hexinlusaurus while it does not in Pulaosaurus and Agilisaurus; the frontals of Pulaosaurus are shorter and more elongated than those of Hexinlusaurus and Agilisaurus; the ventral process of the predentary is notably reduced in Pulaosaurus when compared to that of Agilisaurus; the ossified tendons of Pulaosaurus are in basket-like arrangement of fusiform tendons in caudal region while they are arranged longitudinally in Agilisaurus and Hexinlusaurus; the ungual of manual digit I is sub-conical in Pulaosaurus while it is claw-like in form in Agilisaurus and Hexinlusaurus a supra-acetabular crest is present on the ilium in Pulaosaurus and Agilisaurus while it is absent in Hexinlusaurus; the obturator foramen in the pubis is a notch-like shape in Pulaosaurus while it is a true foramen in Agilisaurus and Hexinlusaurus; the ischial obturator process is near the pubic peduncle in Pulaosaurus while it is located more distally from the pubic peduncle in Agilisaurus and Hexinlusaurus; the ratio between the humerus and the femur in Pulaosaurus is shorter than that of Hexinlusaurus and Agilisaurus; the astragalar ascending process is triangular in Pulaosaurus while it is finger-like in Hexinlusaurus; there are three distal tarsals in Pulaosaurus while there are only two in Agilisaurus and Hexinlusaurus.

Description and comparisons

The specimen is preserved on a brownish-red fine-grained sandstone slab (Fig. 1) and is roughly mediolaterally compressed. The total length of this specimen is approximately 722 mm (i.e., the length from the rostral end of the skull to the caudal end of the last preserved caudal vertebrate) while its skull length is about eight cm from the rostral end of the premaxilla to the caudal border of the parietal and the length of trunk (i.e., the length from the atlas to the caudal border of the pelvic girdle) is about 30 cm. The neurocentral sutures of cervical and caudal vertebrae are not obliterated, which is a possible but not confirmed sign of an immature individual (Bertozzo, Vecchia & Fabbri, 2017). The ratio of the orbit diameter and the skull length is 37%, which is close to the ratio in immature specimens of Jeholosaurus, ranging from 40% to 50% in immature specimens and less than 33% in adult specimens (Barrett & Han, 2009). Because of this, we interpret IVPP V30936 to be an immature individual.

Figure 1 The photograph of the whole skeleton of Pulaosaurus qinglongin left lateral view (IVPP V30936).

Photo credit: Hailong Zhang.

Skull

In lateral view, the skull shape (Figs. 2 and 3, Table 1) is elongate, low, and forms an obtuse triangle. Due to the displacement of the jugal, the exact position of postorbital bones and the shape of infratemporal fenestra are unknown. The supratemporal fenestra is sub-square in lateral view. The orbit is elliptical and is the largest cranial opening with a maximal diameter of which is about three cm. Rostral to the orbit sits the subtriangular antorbital fossa observable in most early-diverging ornithischians including Jeholosaurus, Lesothosaurus, Hexinlusaurus, Agilisaurus (Norman et al., 2004; Barrett & Han, 2009; Peng, 1992; Sereno, 1991; He & Cai, 1984), whose ventral margin is nearly at the same horizon of orbit ventral margin. All of the mandibular elements are disarticulated from each other, so the overall shape of the mandible and the presence or shape of external mandibular fenestra is unknown.

Figure 2 The skull of Pulaosaurus qinglongin left lateral view (IVPP V30936).

(A) Photograph. (B) Outline drawing Abbreviations: Adf, Anterior dentary foramen; Ang, Angular; Art, articular; Ary, arytenoid; A Int, Axial intercentrum; At Int, Atlantal intercentrum; Bo, Basioccipital; Bsp, Basisphenoid; Cera, Ceratobranchial; Cr, Cervical rib; Ecp, Ectopterygoid; Exo, Exoccipital; Fr, Frontal; J, Jugal; Jb, Jugal boss; Lcr, Lacrimal; Ld, Left dentary; Lsp?, Laterosphenoid?; L spl, Left splenial; M, maxilla; Ms, the Meckelian sulcus; Na, Nasal; Nc, Nuchal crest; Nf, Nutrient foramina; Odp, Odontoid process; P, Pterygoid; Pa, parietal; Pal, Palatine; Pa p, Paraoccipital process; Pb, Palpebral; Po, Postorbital; PP, Pterygoid process of basisphenoid; Prd, Predentary; Prf, Prefrontal; Prm, Premaxilla; Sq, Squamosal; Q, Quadrate; Qj, Quadratojugal; Rd, Right dentary; R spl, right splenial; Spo, Supraoccipital; Sur, surangular; 3rd C, the 3rd cervical vertebrate.

Premaxilla (Fig. 2)—Only the left premaxilla is visible. In lateral view, the main body of the premaxilla is subrectangular and possesses two processes: a tapering and caudodorsal-oriented maxillary process that forms the ventral border of the external nares and a reduced nasal process that forms the rostral border of the external nares. The ventral margin of the premaxilla is ventrally offset from the maxilla’s ventral margin as seen in many ornithopods, which is considered a plesiomorphy in neornithischians (Norman et al., 2004). The dorsoventral distance between the premaxillary toothrow and the maxillary toothrow is about two mm. In lateral view, the nasal process of the premaxilla is reduced and does not contact the nasals, possibly due to damage. The maxillary process (i.e., subnarial process) of the premaxilla is large and rostrodorsally contacts the nasal and posteriorly approaches the lacrimal. A premaxilla-lacrimal contact, which is observed in Changmiania (Yang et al., 2020) and Jeholosaurus (Barrett & Han, 2009), may be absent or obscured in Pulaosaurus. The main body of the premaxilla is dorsoventrally concave with its caudal margin being dorsoventrally convex. The lateral surface of the premaxilla’s main body is moderately concave and thickens along the oral margin. Rostral to the first premaxillary tooth, a short edentulous region develops anteriorly into a short beak-shaped process, which is also present in Jeholosaurus (Barrett & Han, 2009), Changchunsaurus (Jin et al., 2010), and Hypsilophodon (Norman et al., 2004). The rostral margin of the premaxillary main body is rugose, suggesting the presence of a rhamphotheca on the premaxilla. This character is considered a plesiomorphy among neornithischians and is observed in Lesothosaurus (Sereno, 1991), Jeholosaurus (Barrett & Han, 2009), Changchunsaurus (Jin et al., 2010), and Hypsilophodon (Norman et al., 2004). The fossa-like depression along the premaxilla-maxilla boundary is absent, whereas it is present in Haya (Barta & Norell, 2021), Orodromeus (Scheetz, 1999), Jeholosaurus (Barrett & Han, 2009) and Changchunsaurus (Jin et al., 2010). There is no arched diastema between the premaxilla and the maxilla, which is a synapomorphy within Heterodontosauridae (Norman et al., 2004). A premaxilla foramen, observed in Haya (Barta & Norell, 2021), Orodromeus (Scheetz, 1999), Jeholosaurus (Barrett & Han, 2009), Changchunsaurus (Jin et al., 2010) and Agilisaurus (Peng, 1992), is also absent. Moreover, there is no prominent narial fossa and the ventral region of the premaxilla does not flare laterally to form the floor of the narial fossa. These morphological characters suggest the premaxilla of Pulaosaurus resembles that of early-diverging neornithischians (Norman et al., 2004).

Figure 3 The photo of several skull elements in lateral view.

(A) The photograph of the visible part of the maxilla. (B) The photograph of the jugal. (C) The line drawing of the jugal. (D) The photograph of the quadratojugal. (E) The photograph of the pterygoid. (F) The line drawing of the pterygoid. (G) The photograph of partial braincase. (H) The line drawing of partial braincase. Photo credit: Hailong Zhang.

Table 1 Measurements of Pulaosaurus qinglong specimen V30936 skull elements.

Skull length	82 mm	
Skull height	42 mm	
Orbit maximal diameter	30.14 mm	
Preorbital skull length	30.84 mm	
Supratemporal fenestra length	15.24 mm	
Frontal length	36.92 mm	
Frontal minimal width	14.60 mm	
Dentary length	39.48 mm	
Right arytenoid length	31.9 mm	
Right arytenoid maximal width	5.8 mm	
Left arytenoid length	34.3 mm	
Left arytenoid maximal width	6.2 mm	
Right ceratobranchial length (visible part)	13.8 mm	
Left ceratobranchial length (visible part)	10.0 mm	

Five bulbous and unserrated premaxillary teeth are present (Figs. 2 and 4), the same number as Haya (Barta & Norell, 2021), Orodromeus (Scheetz, 1999), Changchunsaurus (Jin et al., 2010), but different from the six found in Jeholosaurus (Barrett & Han, 2009), Lesothosaurus (Sereno, 1991) and Agilisaurus (Peng, 1992); four in Convolosaurus marri (Andrzejewski, Winkler & Jacobs, 2019). The premaxillary tooth number of Pulaosaurus is rather primitive compared to later-diverging than tooth counts of most early-diverging taxa within Neornithischia. The first premaxillary tooth is close to the apex of the premaxilla with the second tooth arranged closely to the first. The first premaxilla tooth is smaller than the following four teeth which are all subequal in size to each other. In lateral view, the premaxillary tooth crown is recurved and spade-shaped while the root is long and straight, with the crown moderately expanded mesiodistally and labiolingually above the root. The labial surface of the tooth is smooth, and with no obvious carinae visible. There are wear facets on the distal surfaces of the labial side of the 2nd, 3rd, and 5th premaxillary tooth. The premaxillary tooth row is slightly offset laterally from the maxillary tooth row. No obvious diastema is present between the premaxillary tooth row and the maxillary tooth row.

Figure 4 The teeth of IVPP V30936 under the microscope.

(A) The premaxillary teeth and the most anterior cheek teeth. (B) The anterior cheek teeth. (C) The middle cheek teeth. (D) The posterior cheek teeth. Photo credit: Yunfeng Yang.

Maxilla—The left maxilla is preserved in lateral view (Figs. 2 and 3). It is an elongate, plate-like, trapezoidal bone consisting of a straight, tooth-bearing ramus. The subtriangular ascending process on the rostral end of the maxilla contacts the lacrimal dorsally. The premaxilla overlaps the rostral region of the maxilla, so the ascending process on the rostral end of the maxilla that is observed in many early-diverging neornithischians (Norman et al., 2004; Barrett & Han, 2009; Peng, 1992; Sereno, 1991; He & Cai, 1984) is not fully exposed. The maxilla forms the rostral and ventral borders of the subtriangular antorbital fossa, the dorsal and caudal borders of which are formed by the lacrimal. The maxilla comprises the entire ventral border of the antorbital fossa. Due to taphonomic compression, it is unknown whether additional openings within the antorbital fossa are present or not. It is also unknown whether the anterolateral boss that articulates with the premaxilla is present or not because the rostral region of the maxilla is overlapped by the premaxilla. The maxillary fenestra is absent. There is no slot in the maxilla for the lacrimal. The tooth-bearing ramus mediolaterally widens and the maxillary tooth row is medially inset to form a buccal emargination as seen in most ornithischians (Norman et al., 2004), which is a plesiomorphy in neornithischians (Norman et al., 2004). A ridge extends along the dorsal border of the caudal half of the buccal emargination, starting from the 7th maxillary tooth and extending laterodorsally with its caudal end tapering and flaring laterally. Three or four nutrient foramina lie on the maxillary lateral surface. It is uncertain how many nutrient foramina existed in life due to poor preservation (Figs. 2 and 3).

There are 10 maxillary teeth observed in this specimen, with all preserved in labial view (Figs. 2 and 4). Except for the 8th replacement tooth, the apicobasal and mesiodistal lengths of the teeth increase from the mesially located teeth to the central tooth before decreasing distally. Each maxillary tooth is subtriangular with a distinct neck and a cingulum between the crown and the root, as is the case in many early-diverging ornithischians (Norman et al., 2004; Peng, 1992; Sereno, 1991; He & Cai, 1984). The labial surfaces of the maxillary teeth are relatively smooth with worn faces but no apicobasally extending ridges are present, contrary to Jeholosaurus (Barrett & Han, 2009), Changchunsaurus (Jin et al., 2010), Lesothosaurus (Norman et al., 2004), and Tenontosaurus (Winkler, Murray & Jacobs, 1997). The mesial and distal margins of the teeth are ornamented with simple denticles, which is the plesiomorphic condition that is observed in Jeholosaurus (Barrett & Han, 2009), Lesothosaurus (Norman et al., 2004), and Agilisaurus (Peng, 1992). The numbers of visible marginal denticles from the first to the tenth maxillary tooth are as follows: one denticle on the first tooth; five on the second; seven on the third and the fourth; nine on the fifth; six on the sixth; seven on the seventh and eighth; eight on the ninth and the tenth. The number of visible marginal denticles on the maxillary teeth of Pulaosaurus are apparently influenced by the growth stage of the tooth, the position of each tooth, and the erosion of tooth margin. Broadly, the morphology of the maxillary teeth of Pulaosaurus closely resembles those of early-diverging ornithischians.

Nasal (Fig. 2)—The left nasal is taphonomically compressed with its rostral end broken, so the internarial bar is not preserved. It is unknown whether both nasals were fused in life or not. In lateral view, the left nasal is hatchet-shaped with a tapering, caudodorsally extending caudal process forming the dorsal border of the external nares. The main body of the nasal contacts the maxillary process of the premaxilla ventrally while its caudal process contacts the lacrimal ventrally, overlapping the prefrontal. The contact between the nasal and the frontal is not visible due to the displacement of the bones. In life, the nasals were excluded from the antorbital fossa. No obvious narial fossa (e.g., Jeholosaurus Barrett & Han, 2009) or nasal foramina is observed in the specimen.

Lacrimal (Fig. 2)—In lateral view, the lacrimal is an inverted L-shaped bone that is comprised of a robust rostrodorsal process and the thinner caudoventral process. At the dorsal margin of the lacrimal, the lacrimal contacts the nasal rostrodorsally and the prefrontal caudaldorsally. The rostrodorsal process of the lacrimal contacts the maxilla to form the dorsal border of the antorbital fossa and the caudoventral process of the lacrimal contacts the maxilla to exclude the jugal from the antorbital fossa. The ventral border of the lacrimal overlaps the palatine. The tip of the lacrimal caudoventral process is situated rostral to the jugal and caudodorsal to the maxilla. No lacrimal foramen is visible in lateral view.

Prefrontal (Fig. 2)—In lateral view, the left prefrontal is a rod-like bone which rostrally contacts the lacrimal and the nasal and overlaps the frontal caudally while forming the rostrodorsal margin of the orbit. Rostrally, the prefrontal has a descending ventral process that rostrally articulates with the rod-like palpebral.

Palpebral (Fig. 2)—Among ornithischians, the palpebral is a bone that emanates from the rostrodorsal margin of the orbit and projects across the latter. This element is likely homologous with the anterior supraorbital bone, which refers to one or multiple small osteological elements that are incorporated into the dorsal rim of the orbit (Maidment & Porro, 2010). However, the ornithischian palpebral is not homologous with similar elements reported from other reptilian groups (Nesbitt, Turner & Weinbaum, 2013). The left palpebral is preserved in IVPP V30936. Similar to most ornithischians (Norman et al., 2004), the palpebral of Pulaosaurus is a rod-like bone with its rostral end articulating with the prefrontal. In IVPP V30936, the caudal end of the palpebral is overlapped by the displaced jugal. The length of the palpebral is more than 80% of the orbit’s diameter and its caudal end projects into the orbit freely, which is a plesiomorphy of neornithischians (Han et al., 2018). The palpebral does not articulate with the postorbital as seen in Agilisaurus (Peng, 1992; He & Cai, 1984). There is no indication of a postpalpebral observable in Haya (Barta & Norell, 2021) or Thescelosaurus (Boyd, 2014).

Jugal (Figs. 2 and 3)—The left jugal is preserved, though it is taphonomically displaced. In lateral view, the jugal is a plate-like bone composed of three rami and it forms the caudoventral border of the orbit and the anteroventral border of the infratemporal fenestra. Due to its displacement, its points of contact with other bones remains uncertain. In contrast with Yinlong (Han et al., 2018), the jugal is excluded from the antorbital fossa as observed in Jeholosaurus (Barrett & Han, 2009), Haya (Barta & Norell, 2021), Hexinlusaurus (He & Cai, 1984), and Agilisaurus (Peng, 1992). This condition is a plesiomorphy in neornithischians where the jugal is excluded from the antorbital fossa. The rostral process is in rostroventrally oriented and the dorsal process is in caudaldorsally oriented. Due to displacement, the jugal’s articulating surface with the postorbital is exposed and the jugal is articulated with the postorbital with a ‘finger-in-recess’ joint. There is a small boss on the caudalventral corner of the jugal dorsal process. The boss is similar to, but much smaller than, the ones found in Orodromeus (Scheetz, 1999). Similar ornaments are also present in Orodromeus (Scheetz, 1999), Heterodontosaurus (Jin et al., 2010), Zephyrosaurus (Jin et al., 2010), and marginocephalians (Jin et al., 2010) but absent in Ornithopoda (Jin et al., 2010). However, the boss on the jugal of Pulaosaurus is located on the dorsal process of the jugal while the jugal boss is located on the ventral margin of the jugal in Orodromeus (Scheetz, 1999), Heterodontosaurus (Jin et al., 2010), and Zephyrosaurus (Jin et al., 2010). The homology of these ornaments in different taxa is questionable and beyond the scope of this project. It is possible that the size of the jugal boss is ontogenetically variable as is the case in Orodromeus (Scheetz, 1999). The dorsal branch of the caudal process is weakly expanded caudodorsally while the lower, dorsoventrally wider branch expands caudoventrally. A bifurcated caudal process is a synapomorphy present in Lesothosaurus (Norman et al., 2004), Jeholosaurus (Barrett & Han, 2009), Changchunsaurus (Jin et al., 2010), and psittacosaurs (You & Dodson, 2004) but is absent in many early-diverging cerapodans including pachycephalosaurs (Jin et al., 2010) and Yinlong (Jin et al., 2010). There are no ornamentations on the lateral side of the jugal, which contrasts what is observed in Jeholosaurus (Barrett & Han, 2009).

Frontal (Fig. 2)—In lateral view, the left frontal is an elongated, narrow, and forms an arched shelf that comprises the rostrodorsal roof of the skull. This character is a synapomorphy in early-diverging neornithischians (Barrett & Han, 2009). The rostrodorsal margin articulates with the prefrontal, the rostroventral corner articulates with the palpebral, and it caudally contacts the parietal. The contact between the frontal and the nasal is not visible due to the taphonomic displacement of the bones and the frontal contacts the postorbital caudoventrally. Although the exact ratio between frontal’s length and width is unknown, the value is estimated to be about 2.5. The estimated length and width ratio is smaller than that of Jeholosaurus (Barrett & Han, 2009), Agilisaurus (Peng, 1992), Hypsilophodon (Barrett & Han, 2009), and Zephyrosaurus (Jin et al., 2010) but greater than that of early-diverging ceratopsians (Barrett & Han, 2009), Hexinlusaurus (Barrett & Han, 2009), Orodromeus (Scheetz, 1999) and Thescelosaurus (Barta & Norell, 2021; Barrett & Han, 2009). The lateral margin thins and flares laterally, thus forming the orbital margin. The orbital margin is not as rugose as other ornithischians. There exists a groove on the rostrodorsal margin of the orbital margin that articulates with the prefrontal and curves ventrally. The dorsal margin of the frontal thins and projects caudodorsally, which makes the dorsal margin convex and where it reaches its highest point over the orbit. Between the frontal and the parietal, the suture is preserved as a short scarf-joint that is located behind the orbit. Caudally, the margin of the frontal is overlapped by the parietal.

Postorbital (Fig. 2)—In lateral view, the postorbital is a triradiate-shaped bone that is composed of an infratemporal process that extends rostrodorsally, a jugal process that extends rostroventrally, and a squamosal process that extends caudally. The postorbital and the squamosal form the temporal bar, the former of which contributes more. The surface of IVPP V30936’s postorbital is smooth and lacks ornaments on its surface while ornamentation on the postorbital is a synapomorphy of Jeholosaurus (Barrett & Han, 2009), Haya (Barta & Norell, 2021), and Orodromeus (Scheetz, 1999). The orbital margin of the postorbital has no protuberance projecting into orbit as observed in Haya (Barta & Norell, 2021) and Orodromeus (Scheetz, 1999). Rostrally, the end of the infratemporal process is broken from the infratemporal process. However, based on the length of the infratemporal process and its broken rostral end, it is assumed that the length of the infratemporal process is about 11.4 mm while the length of the squamosal process is about 7.5 mm. The infratemporal process is articulated with the caudoventral corner of the frontal. The squamosal process tapers into a narrow tip where it articulates with the squamosal ventrally. In IVPP V30936, the length of the squamosal process suggests that the rostrocadual length of infratemporal fenestrae is relatively short. The jugal process is overlapped by the quadrate, so its specific morphological characters and specific contact with the jugal remain unknown. However, based on the morphology of the jugal’s dorsal process, the rostrocadual width of the jugal process of the postorbital decreases, ends on the orbital margin and it articulates with the jugal with a ‘finger-in-recess’ joint.

Parietal (Fig. 2)—In lateral view, the left parietal is a saddle-shaped bone. Due to compression and displacement, the original contacts between the parietal and other cranial elements are unknown, so the description is based on the observable contacts between cranial elements. The parietal forms the caudodorsal roof of the skull, the infratemporal fenestrae, and part of the supratemporal fenestrae. Rostrally, the parietal contacts the frontal along with the postorbitals and squamosals laterally. The caudal margin of the parietal flares laterally to form the nuchal crest as isA morphological feature is observed in Haya (Barta & Norell, 2021), Changmiania (Yang et al., 2020), Orodromeus (Scheetz, 1999), and Oryctodromeus (where the nuchal crest is formed by the supraoccipital) (Krumenacker, 2017) and considered as a adaptative characteristic for digging. Along the supratemporal fenestrae, the parietal forms the caudal margin. It is unknown whether a sagittal crest or a median process inserts between the frontals.

Squamosal (Fig. 2)—In lateral view, although overlapped by the squamosal process of the postorbital, the squamosal is a small and triradiate bone; however, the quadrate process has been taphonomically broken away. The dorsal surface of the squamosal is smooth and expands laterally. In rostrolateral view, the quadrate process is observable and its lateral surface forms a glenoid on the quadrate process where the squamosal articulates with the quadrate, as is observed in early-diverging ceratopsians such as Yinlong (You & Dodson, 2004), Hualianceratops (You & Dodson, 2004), and Liaoceratops (You & Dodson, 2004). The contacting relationships between the squamosal and other bones remain uncertain due to the displacement of bones.

Quadrate (Fig. 2)—The left quadrate is exposed in lateral view. Two-thirds of the upper quadrate shaft is bowed rostrally. The proximal head of the quadrate is bluntly round. Distally, the condyles are overlain by the displaced pterygoid while the quadrate articulates with the squamosal proximally. Due to displacement, the exact contact relationships between the quadrate and other cranial bones are unknown. The shaft of the quadrate flares laterally to form the rostrolateral-oriented jugal wing of the quadrate, which arises from the dorsal margin and terminates just above the distal condyle. The pterygoid wing and the distal condyle are overlapped, so their morphological characteristics are unknown. However, it is hypothesized that the quadrate condyle is ventrally offset from the level of the maxillary tooth row based on the position of the surangular retroarticular process. It could not be determined whether a quadrate foramen is present or not.

Quadratojugal (Figs. 2 and 3)—In lateral view, the left quadratojugal is a subtriangular and flat, thought displaced, bone. Due to displacement, the relationships between the quadratojugal and other cranial elements is unknown. Also, displacement has caused the quadratojugal to be partially overlapped by the jugal with its ventral margin covered by the ectopterygoid. The quadratojugal is comprised of an elongated process, a dorsoventrally thick process and a third process whose shape is uncertain. The estimated shape of the overlapped process is drawn with a dashed line. Due to displacement of the quadratojugal and overlapping of the quadrate, the exact lengths and orientations of the processes are unknown. It cannot be determined whether a quadratojugal foramen is present as is observed in neornithischians such as Hypsilophodon (Norman et al., 2004), Jeholosaurus (Barrett & Han, 2009).

Pterygoid (Figs. 2 and 3)—In lateral view, the medial side of the right pterygoid is exposed due to taphonomic displacement. Anatomically, the pterygoid consists of three processes: the quadrate process, the mandibular process, and the palatine process. The quadrate process is a thin, subtriangular sheet that projects caudodorsally and is comparably much smaller than the same process of Thescelosaurus (Boyd, 2014). Medially, a caudomedially facing cup-shape that receives the basipterygoid process of the basisphenoid arises from the rostral corner of the quadrate process, where it joins with the other processes. The mandibular process is a thin, small, rostrocaudally elongate, and subtriangular process that projects dorsally,arising from the caudal corner of the quadrate process. Finally, the palatine process originates from the caudomedially facing cup-shape. The palatine process extends and then tapers rostrally and it forms a lateromedially oriented tab that contacts the palatine. This tab is much shorter relative to the same tab found on the pterygoid’s palatine process of Thescelosaurus (Boyd, 2014).

Ectopterygoid (Fig. 2)—In lateral view, the ectopterygoid is a small and caudally bowed bar with a notch on the dorsal margin, similar to that of Thescelosaurus (Boyd, 2014). The ectopterygoid extends caudoventrally where it is close to the caudal end of the maxilla. A notch located along the ectopterygoid is interpreted as the postpalatine fenestra. Due to taphonomic displacement, the exact contact relationships between the ectopterygoid and other bones remain uncertain, but it is possible that the ectopterygoid contacted the palatine rostrally.

Palatine (Fig. 2)—In lateral view, the palatine is a sheet-like bone whose caudal end extends laterally, forming a pyramidal boss that likely represents the point of articulation for the pterygoid. Laterally, the palatine is overlapped by the lacrimal and the maxilla. Only the caudal end of the palatine is observable within the orbit. Due to overlapping and mediolateral compression, the overall morphology of the palatine and its contact relationships with other adjacent bones remains uncertain.

Laterosphenoid(?) (Fig. 2)—In lateral view, a small, laterally extending, pyramidal protuberance lies rostroroventrally to the orbital process of the postorbital. Based on its relative location, it is possible that it could be part of the braincase and is interpreted as the laterosphenoid. This protuberance is similar to the lateral articular head of the laterosphenoid that would articulate with a socket on the ventral margin of the frontal. However, due to poor preservation and overlapping, the true identification of this element, its morphology, and therelationships between this element and other bones remain indeterminate.

Basioccipital (Figs. 2 and 3)—The basioccipital of IVPP V30936 is disarticulated from the supraoccipital and is preserved upside down relative to its life position next to the basisphenoid. Rostroventrally, the basioccipital contacts the basisphenoid, but the suture between the two elements is ambiguous. In lateral view, the basioccipital is an inverted saddle-shaped element, as is the case in many neornithischians (Barta & Norell, 2021; Jin et al., 2010; Peng, 1992; Scheetz, 1999). The basioccipital lies rostrodorsal to the axis and caudoventral to the parietal, with its rostrooventral margin overlapped by the atlas and its rostroodorsal margin. The caudal end of the basioccipital narrows dorsoventrally to form the caudally oriented occipital condyle. The shape of the occipital condyle is unknown as it is only visible in lateral view. It is also unknown how much the basioccipital contributes to the condyle or whether only the basioccipital contributes to the condyle. The rostral region of the basioccipital projects slightly ventrally and becomes dorsoventrally thicker, which then forms the caudal part of the basal tuber. As the ventral side of the basal tuber is not visible, the exact shape of the basal tuber cannot be confirmed. However, based on the ventral margin of the basioccipital, it is assumed that the basal tuber is undeveloped and the length of the basal tuber is about two cm.

Basisphenoid (Figs. 2 and 3)—As described above, the basisphenoid is preserved upside down and the suture between the caudal basisphenoid and the rostral basioccipital is ambiguous. In total, the length of the basisphenoid is subequal to the length of the basioccipital. The main body of the basisphenoid is visible in the lateral view, and it has the shape of a rostrodorsally oriented cubic forming the rostral part of the basal tuber. No foramina or openings are visible on the lateral side of the basisphenoid. Its rostral region tapers rostrally to support the bifurcated basipterygoid processes, which are broken away but are visible ventrally. Both basipterygoid processes are rostroventrally oriented and blunt at their ends, characteristics that are considered plesiomorphies in neornithischians (Barta & Norell, 2021; Peng, 1992; Scheetz, 1999). The angle between the two basipterygoid processes and the main body is approximately 35° degrees.

Supraoccipital (Fig. 2)—In lateral view, the supraoccipital is a sheet-like bone that is situated caudally to the parietal. Due to compression, there is little information about its morphological characters and contact relationships. It is unknown how much or even whether the supraoccipital contributes to the magnum foramen.

Exoccipital (Figs. 2 and 3)—Only a portion of the left exoccipital is observable in medial view. This partial exoccipital is preserved upside down with its paroccipital process taphonomically damaged. The exoccipital contacts the occipital condyle contacts the occipital condyle ventrally and forms the dorsolateral region of the foramen magnum. The rostrodorsal border of the exoccipital extends medially to form the articular surface for the supraoccipital, so it is hypothesized that the supraoccipital contributes to the dorsal margin of the magnum foramen. There is a canal ventral to the articular surface for the supraoccipital, but the function of the canal is difficult to determine due to poor preservation. The preserved portion of the paroccipital process is short and blunt. On its caudodorsal corner, the exoccipital extends moderately laterally.

Predentary (Fig. 2)—The left predentary is in close association with the dentary and is only visible in lateral view as a mediolaterally and rostrocaudally compressed element. In lateral view, the predentary is triangular with a short ventral process. It is unknown whether the ventral process is forked or not. The rostrocaudal length of the predentary is short at approximately half the rostrocaudal length of the premaxilla, which is a plesiomorphy among neornithischians (Butler, Upchurch & Norman, 2008). The dorsal margin of the predentary is longer than the ventral margin. The rostral tip does not project above the main body, similar to Haya (Barta & Norell, 2021) and Changchunsaurus (Jin et al., 2010). The ventral process is shorter than the lateral process.

Dentary (Fig. 2)—Both dentaries are preserved and observable. The left dentary is visible in lateral view and the right dentary is observable in medial view. Because the postdentary elements are displaced, the exact relationships between the dentary and other elements are uncertain. The dentary is an elongated and dorsoventrally thin element with a prominent, caudodorsally oriented process on the caudal end that contributes to the coronoid process. The dorsal and ventral margins of the dentary are subparallel to each other along most of the dentary length with the dorsal margin being greater in length than the ventral margin. Both margins converge rostrally to form the articulate surface for the predentary, which is located at the midpoint of the dentary in height, as observed in Changchunsaurus (Jin et al., 2010). Rostral on the dentary, a vascular from the anterior dentary foramen forms an impressed canal extending rostrally to the predentary. The lateral side of the dentary is smooth and unornamented. Four nutrient foramina are present on the lateral side and are located near the dorsal margin of the dentary (Fig. 2). The caudoventral region of the dentary is concave and forms the articulation for the surangular and the angular. This articulation extends for one-third the length of the dentary along the dentary’s caudal borderand extends to the midpoint of the dentary in height. The dentary reaches its maximal height on its caudal margin, forming a caudodorsally projecting process that contributes to the coronoid process. The medial surface of the dentary is relatively flat mediolaterally when compared to the lateral side. The Meckelian sulcus is located near the ventral margin along the medial surface of the dentary. The Meckelian sulcus originates from the rostral end of the dentary but it is unknown how far it extends caudally. The caudoventral margin of medial dentary surface is covered by the plate-like splenial.

The dentary tooth row is medially inset to form the buccal emargination (Fig. 4), similar to the maxillary tooth row. Because the dentary tooth row is obscured by the maxillary tooth row, the exact number of dentary teeth is unknown. The toothrow stops at the base of the coronoid process. Only six dentary teeth are visible between the rostral and caudal ends of the left dentary. The shape of dentary teeth resembles the maxillary teeth. The first small dentary tooth is adjacent to the predentary, similar to many early-diverging neornithischians (Norman et al., 2004; Sereno, 1991; He & Cai, 1984).

Splenial (Fig. 2)—Both splenials are preserved although the rostroventral end is missing because of a taphonomical fracture of the dorsal border of the right splenial. Most of the left splenial is overlapped by the dentary, though the right splenial is observable in medial view. Although their exact contacting relationships with other bones are ambiguous due to displacement, the preservation state suggests that they cover the caudoventral margins of the dentary’s medial margins. The splenial is a subtriangular sheet. Its caudal end bifurcates into a blunt dorsal process and an elongate, tapering caudoventral process. From rostral to caudal, the dorsoventral length of the right splenial thickens. The rostroroventral end of the right splenial is broken and lost, so it is uncertain whether there exists a foramen on the rostral end.

Surangular (Fig. 2)—The left surangular is displaced and preserved upside down, thus exposing its medial side. It is a roughly flat, subtriangular element that is composed of three processes in medial view with no observable foramina along its surface. There are no foramina observed in other neornithischians such as Haya (Barta & Norell, 2021), Thescelosaurus (Boyd, 2014), and Changchunsaurus (Jin et al., 2010). Due to taphonomic displacement, the exact contacting relationships with other elements is unknown. The rostrodorsal process of the surangular is an elongate, arched, and slender process that possibly contacts the angular ventrally. At its base, the medial process of the surangular is flat. The rostroventral process is short and blunt-ended, forming the ventral border where the surangular meets the angular. The elongated, caudal-oriented process on the caudoventral corner is the retroarticular process, whose caudal end becomes concave to contact the articular.

Articular (Fig. 2)—The articular is a subtrapezoidal element when viewed laterally. It is disarticulated from the retroarticular process and the quadrate, so its specific relationships with other elements is indiscernible. The articular surface for the quadrate is not visible. The ventral border of the articular extends laterally to form a ridge.

Angular (Fig. 2)—The left angular is visible in lateral view and forms the caudoventral margin of the mandible. It is a subrectangular element, and, asother displaced cranial elements, the contact relationships of this bone with other elements remain uncertain. On its rostral border, the angular is mediolaterally concave to articulate with the dentary. The caudal border is mediolaterally convex and forms the possible articulating surface for the surangular.

Hyolaryngeal apparatus (Fig. 5, Table 1)—Two arytenoids are elongate, flat, and leaf-shaped with a L-shaped cross-section and are located ventral to the mandible. The length of the arytenoids is about 80% of the dentary length. This pair of elements is similar to ossified arytenoids observed in Pinacosaurus (IGM100/3186) (Yoshida, Kobayashi & Norell, 2023). Each arytenoid is composed of a mediodorsal wing and a dorsolateral wing. The mediodorsal wing projects mediodorsally to form the arytenoid process which serves as an attachment site for the M. dilator (Yoshida, Kobayashi & Norell, 2023). The dorsolateral wing widens caudally before tapering at its caudal end. The lateral edge, especially at its rostral end, is rugose, which is the possible articulation for the cricoid.

Figure 5 Arytenoids and ceratobranchials of Pulaosaurus qinglong in laterodorsal view (IVPP V30936).

(A) The photograph of ceratobranchials and arytenoids. (B) The line drawing of ceratobranchials and arytenoids. Abbreviations: ap, arytenoid process; c, ceratobranchial; lw, laterodorsal wings of the arytenoids; mw, mediodorsal wings of the arytenoids. Photo credit: Hailong Zhang.

A pair of elongated, gracile rod-shaped elements (Fig. 5)—which are interpreted here to be the ceratobranchials—are overlapped by the arytenoids and meet at a point that is caudooventral to the mandible. The length of ceratobranchials is less than 40% of the dentary length.

Postcranial elements

Axial skeleton

Cervical series (Figs. 2 and 6, Table 2)—A mostly complete cervical series is preserved in the specimen. Due to deformation and obscured elements, the exact number of cervical vertebrae is unknown. However, nine cervical vertebrae are assumed to be preserved in the specimen, since nine cervical vertebrae usually occur in early-diverging ornithopods (Han et al., 2012), while fewer occur in Thyreophora (Han et al., 2012) and more than nine cervical vertebrae appear in later-diverging ornithopods and ceratopsians (Han et al., 2012). The first three cervical vertebrae are visible in the lateral view with the following vertebrae being partially observable in dorsal view, though they are notably deformed and obscured. The 8th and 9th cervical vertebrae are disarticulated from the rest of the series. Most of the cervical ribs are disarticulated and scattered with the exception of the 5th and 6th cervical ribs. The sutures of the axis and 3rd cervical vertebra between the cervical neural arches and centra are unfused.

Figure 6 Cervical series of Pulaosaurus qinglong: (IVPP V30936). Atlas, axis and 3rd cervical visible in left lateral view; from 4th cervical to 9th cervical visible in dorsal view.

(A) The photograph of the cervical series. (B) The line drawing of the cervical series. Abbreviation: a int, axial intercentrum; cr, cervical rib; dia, diapophysis; ns, neural spine; od p, odontoid process; pa, parapophysis; pop, postzygapophysis; tp, transverse process. Photo credit: Hailong Zhang.

Table 2 Measurements of Pulaosaurus qinglong specimen V30936 skull elements.

Right coracoid length in medial view	19.86 mm	
Right coracoid maximal width in medial view	16.68 mm	
Left coracoid length in lateral view	18.52 mm	
Left coracoid maximal width in lateral view	?	

The atlas (Figs. 2 and 6) is visible in left lateral view. The proatlas and the atlantal ribs are not preserved. The atlantal intercentrum is disarticulated from the neural arches and is visible in rostroventral view, ventral to the axis. In ventral view, the atlantal intercentrum is subcrescentic in shape. The anterior side of the atlas intercentrum is excavated by a fossa that is the deepest on the midline and becomes shallower laterally. The lateral margins of the atlas intercentrum expand dorsoventrally to form the transverse processes to articulate with the atlantal ribs. A flange extends on the ventral margin that demarcates the anterior side of the atlas intercentrum from the ventral side. A groove is located along the ventral side of the atlas, similar to what is observed in Jeholosaurus (Han et al., 2012) and Hexinlusaurus (He & Cai, 1984). In lateral view, the neural arches are a pair of wing-shape elements that expand dorsolaterally and are jointed at the base. The transverse width of the base narrows from rostral to caudal. The base has two articulating surfaces, a caudodorsal one for the odontoid process and a ventral one for the intercentrum. Though taphonomically compressed, the odontoid process is caudal to the atlas. The prezygapophysis is not visible while the postzygapophysis is developed to articulate with the prezygapophysis of the axis.

The axis (Figs. 2 and 6) is well-preserved, is observable in lateral view, and is comprised of the intercentrum, the centrum and the neural arch. Dorsally, the neural spine forms a dorsocaudally oriented crest. Its relative rostrocaudal length is not as long as those of many neornithischians in which it extends beyond the caudal margin of the third cervical (Barta & Norell, 2021; Han et al., 2012; Butler et al., 2011). At the base of the neural spine, the prezygapophysis is located on the rostral end and a well-developed postzygapophysis on the caudal end that articulates with the prezygapophysis of the third cervical. A round diapophysis is located ventral to the prezygapophysis on the rostrolateral surface of the neural arch. The height of the neural arch is subequal to that of the centrum. The suture between the neural arch and the centrum is not fused. In lateral view, the centrum is opisthocoelous. The paraphysis is a small process present on the rostrodorsal surface of the centrum. A wedge-shaped intercentrum is rostralto the centrum. The axial rib is taphonomically disarticulated and missing. Based on the presence of both the diapophysis and the parapophysis, the axial rib was likely double-head.

The 3rd cervical (Fig. 6) is observable in lateral view though its caudal region is broken due to poor preservation. The overall shape and composition of the third cervical vertebra is similar to that of the axis except it lacks an intercentrum. Compared to the axis, the neural spine of the third cervical is much smaller and more caudally oriented. The prezygopaphysis is better developed and extends rostrally over the caudal margin of the axis centrum. The diapophysis of the 3rd cervical is smaller and more caudalventrally located. When compared with the parapophysis of the axis, the parapophysis of the 3rd cervical is larger.

The following cervical series (Fig. 6) are visible in dorsal view but are notably damaged and obscured. Their rostrocaudal length is longer than their transverse width. The 4th cervical vertebra is composed of a pair of visible, well-developed, laterocaudally oriented transverse processes, well-developed postzygapophyses, and a small, dorsocaudally oriented neural arch. The possible two-headed rib of the 4th cervical vertebra is ventral to the transverse process. The 5th cervical vertebra is comprised of a pair of well-developed prezygapophyses, a laterocaudally oriented transverse processes that articulates with a double-headed rib, and a small, dorsocaudally oriented neural arch. The 6th cervical vertebra is comprised of a small, dorsocaudally oriented neural arch, and a laterocaudally oriented transverse process that articulates with a double-headed rib with more than twice the rostrocaudal length of the centrum of the 6th cervical vertebra. The 7th, 8th and 9th cervical vertebrae are so compressed that little morphological information can be meaningfully extracted from them.

Dorsal vertebrae and dorsal ribs (Fig. 7)—Most dorsal vertebrae are obscured, deformed, or overlapped by the ribs and ossified tendons. Only four distal dorsal vertebrae close to the ilium can be observed in lateral view although these dorsal vertebrae are deformed. The dorsal vertebrae are amphiplatyan or moderately procoelous. Only one obvious diapophysis on one dorsal vertebrate centrum is visible. There is no unambiguous evidence of striated rims around the rostrolateral and caudolateral borders of the centra. Most dorsal ribs are obscured or disarticulated, and all are compressed. Six recognizable double-head and moderately bowed dorsal ribs are articulated with the proximal dorsal vertebrae (Fig. 7). There is no evidence of the existence of intercostal processes.

Figure 7 The visible dorsal vertebrae and the ossified tendons in lateral view.

(A) Dorsal ribs and ossified tendons in lateral view. (B) The photograph of four visible dorsal vertebrae. (C) The line drawing of four visible dorsal vertebrae. Abbreviations: C, Centrum; Dia, Diapophysis; NA, the Neural Arch; NS, the Neural Spine; Post, Postzygapophysis. Photo credit: Hailong Zhang.

Sacral vertebrae—All sacral vertebrae are overlapped by the ilium.

Caudal vertebrae—In lateral view, much of the caudal series of vertebrae is preserved. The preserved caudal vertebrae include two anterior caudal vertebrae (Fig. 8A), five middle caudal vertebrae (Fig. 8B), and 17 posterior caudal vertebrae (Fig. 8C). Most anterior caudals are damaged or missing while most of the posterior vertebrae are missing. All preserved caudal vertebrae are amphiplatyan. The anteroposteriorlength of the caudal vertebrae gradually increases while their dorsoventral height decreases. The neurocentral sutures of all caudal vertebrae are unfused.

Figure 8 Caudal vertebrae series of Pulaosaurus qinglong in left lateral view (IVPP V30936).

(A) The proximal caudal vertebrae. (B) The middle caudal vertebrae. (C) The distal caudal vertebrae. Photo credit: Hailong Zhang.

The centra of the anterior caudals are dorsoventrally concave. The ratio between the anteroposterior length and dorsoventral length of the vertebrae is approximately 1.5. Their neural arches are low and all neurocentral sutures are unfused. Both prezygapophyses and postzygapophyses extend a little beyond the anterior and posterior margins of the centrum, respectively. The hatchet-shaped neural spines are flat and more elongate than those of the posterior caudals, are positioned moderately anterior to the postzygapophyses extending posterodorsally, and are dorsoventrally expanded. The chevrons are disarticulated and are an elongated rod-shape in lateral view.

The middle caudal vertebrae are similar to the anterior caudals except that the dorsoventral length of the middle caudal centrum is shorter than that of anterior ones. Small transverse processes are present at the base of the neural arches. In lateral view, there are two hatchet-shaped chevrons that are disarticulated with the centrum.

Proximodistally, the dorsoventral length of the posterior caudal vertebrae decreases and the anteroposterior length increases to be more than twice the dorsoventral length. The prezygapophyses extend beyond the anterior end of the centrum and the postzygapophyses do not extend beyond the posterior end. Anteroposteriorly, the anteroposterior length of the prezygapophyses and postzygapophyses gradually increases, the dorsoventral lengths decrease, and both are fused into a single, unified structure. There are fourteen chevrons of the posterior caudal vertebrae preserved, twelve of which are articulated with the anteroventral corner of the centrum whose orientations gradually shift from vertical to horizontal in relation to the centrums. In lateral view, the chevrons are flat and hatchet-shaped while in anterior view they are ‘Y’-shaped with two gracile proximal processes merging into one ventrally oriented tapering process.

Ossified tendons (Fig. 7A)—Ossified tendons are present along with the vertebral column. They are found alongside the middle dorsal vertebrae and the sacral vertebrae. The ossified epaxial tendons present on caudal vertebrae are not visible in this specimen and it is possible that the ossified tendons are absent from the caudal vertebrae. In lateral view, the ossified tendons are preserved as compressed, black filaments that are arranged in a basket-like arrangement of fusiform tendons along the caudal region. They lack a lattice arrangement and are more similar to a linear bundle arrangement.

Appendicular skeleton

Scapula (Fig. 9)—Both scapulae are heavily damaged and compressed. However, it is apparent that the scapulae formed an elongate and strap-like shape in life. Only part of the proximal plate of the right scapula, and fragments of the proximal and distal ends of the left scapula are observable in lateral view. As a result, the exact shape of either scapula remains indeterminate. The morphology and position of the glenoid are also unknown. Similar to most small ornithischians (Han et al., 2018), but different from Koreanosaurus (Min et al., 2011) and Oryctodromeus (Krumenacker, 2017), the scapula is unfused with the coracoid. The acromion, positioned at the rostrodorsal corner of proximal plate, is blunt, not prominent, and does not develop into the scapular spine. Both the dorsal and the ventral margins of the proximal end of the scapula are concave. A laterally concave fragment of the distal end is located far from the proximal plate. Although the overall shape of the distal end of the scapula is unknown due to damage, the fragmentary remains suggest that the dorsoventral width of the scapula gradually decreases caudally.

Figure 9 The pectoral girdle of Pulaosaurus qinglong (IVPP V30936) in lateral view and the sternum of Pulaosaurus qinglong (IVPP V30936) in ventral.

Photo credit: Hailong Zhang.

Coracoid (Fig. 9 and Table 3)—Both coracoids are preserved but displaced. The description of the orientation is based on the fully exposed right coracoid. In lateral view, the coracoid is a smooth, subquadrate plate with a concave surface. The dorsoventral length is subequal to the rostrocaudal length. The rostral border of the coracoid is relatively straight while the caudal border is strongly concave and forms an embayment that is observed in many neornithischian species (Barta & Norell, 2021; Min et al., 2011; He & Cai, 1984; Yang et al., 2020). The dorsal border, which is sutured with a scapula, is rugose while the ventral border is smooth. Unlike Haya (Barta & Norell, 2021), Hexinlusaurus (He & Cai, 1984), Jeholosaurus (Han et al., 2012), and many other taxa, no enclosed foramen is present laterally along the coracoid. Based on the rugose morphology of the dorsal border, the coracoid foramen could be on the dorsal border.

Table 3 Measurements of Pulaosaurus qinglong specimen V30936 cervical series.

Axial length	8.50 mm	
3rd cervical centrum length	9.04 mm	
4th cervical centrum length	12.76 mm	
5th cervical centrum length	?	
6th cervical centrum length	15.62 mm	
7th cervical centrum length	?	
8th cervical centrum length	?	
9th cervical centrum length	?	

Sternum (Fig. 9)—Only a proximal fragment of the left sternum is preserved. The degree of fracturing to the sternal elements has made the total shape indeterminable. The proximal fragment is mediolaterally thin. This fragment is fan-shaped along its proximal end and is dorsoventrally broad but gradually narrows from rostrocaudally, so it is likely that the sternum is a hatchet-shaped or a shafted element.

Humerus (Fig. 10, Table 4)—Only the left humerus is preserved completely with its head partially overlapped by the coracoid. The left humerus is visible in lateral view. Generally, the humerus is similar to many small early-diverging neornithischians (Han et al., 2018), and it is a relatively gracile element compared to the humerus of Koreanosaurus (Min et al., 2011) and Tenontosaurus (Winkler, Murray & Jacobs, 1997). The length of the humerus is about half the length of the femur. The rostrocaudally thin proximal end of the humerus expands mediolaterally to form a centered head, which is rotated 37° relative to the shaft. The anterior surface of the head has a concave shape that forms the bicipital sulcus. The deltopectoral crest is damaged, so its total shape is poorly understood. The lateral tuberosity is obscured by the coracoid while the medial tuberosity is continuous with a ridge defining the proximal end concavity. The shaft is gracile and short with an elliptical cross-section. An intercondylar groove extends along the shaft to form an oval intercondylar fossa near the distal end of the humerus. The ulnar condyle is larger than the radial condyle and extends anterodistally.

Figure 10 The left humerus in view.

Photo credit: Hailong Zhang.

Table 4 Measurements of Pulaosaurus qinglong V30936 forelimbs.

Left humerus length in anterior view	50.16 mm	
Distal end width of left humerus in anterior view	10.86 mm	
Proximal end width of left humerus in anterior view	?	
Shaft minimal width of left humerus in anterior view	6.04 mm	
Left ulna length in anterior view	40.86 mm	
Distal end width of left ulna in anterior view	7.30 mm	
Proximal end width of left ulna in anterior view	10.12 mm	
Shaft minimal width of left ulna in anterior view	4.92 mm	
Left radius length in anterior view	39.24 mm	
Distal end width of left radius in anterior view	6.76 mm	
Proximal end width of left radius in anterior view	6.24 mm	
Shaft minimal width of left radius in anterior view	3.40 mm	
Right ulna length in posterior view	43.24 mm	
Distal end width of right ulna in posterior view	5.4 mm	
Proximal end width of right ulna in posterior view	9.24 mm	
Shaft minimal width of right ulna in posterior view	4.04 mm	
Right radius length in posterior view	41.8 mm	
Distal end width of right radius in posterior view	6.7 mm	
Proximal end width of right radius in posterior view	5.48 mm	
Right radius shaft minimal width in posterior view	3.58 mm	
Left manual portion length	32.5 mm	
Longest left manual digit(2nd digit) length	29.76 mm	
Left metacarpal I length	7.62 mm	
Left metacarpal II length	13.08 mm	
Left metacarpal III length	12.19 mm	
Left metacarpal IV length	8 mm	
Left metacarpal V length	5.36 mm	

Ulna and radius (Fig. 11, Table 4)—The right and left ulnae, and right and left radii are all preserved in lateral view. Both ulnae and radii are straight, rod-like elements, both shorter than the humerus. The proximal end of the ulna is broad, and the shaft narrows distally with the distal end moderately expanded. The olecranon process of the ulna is low, which is a plesiomorphy usually observed in early-diverging neornithischians (Peng, 1992; He & Cai, 1984). Laterally, the border of ulna’s proximal end is keeled, as is observed in Orodromeus (Scheetz, 1999), Changmiania (Yang et al., 2020) and Koreanosaurus (Min et al., 2011). The proximal ulna has a concave and triangular articular facet for articulation with the radius. Distally, the ulna has a similar morphology and articular facet for articulation with the radius. However, the distal articular facet is smaller and shallower than the proximal one. Compared to the ulna, the radius is slender and gracile. The shaft of the radius is about 30% narrower than the ulna shaft (Table 4).

Figure 11 The left ulna, the left radius in anterior view and the left manus in view.

Abbreviations: DC, Distal Carpal; Int, Intermedium; R, Radiale; U, Ulnare; Mc I, Metacarpal I; Mc II, Metacarpal II; Mc III, Metacarpal III; Mc IV, Metacarpal IV; Mc V, Metacarpal V; I-1, Phalanx I-1; I-2, Phalanx I-2; II-1, Phalanx II-1; II-2, Phalanx II-1; II-3, Phalanx II-3; III-1, Phalanx III-1; III-2, Phalanx III-2; III-3, Phalanx III-3; III-4, Phalanx III-4; IV-1, Phalanx IV-1; IV-2, Phalanx IV-2; IV-3, Phalanx IV-3; V-1, Phalanx V-1; V-2, Phalanx V-2. Photo credit: Hailong Zhang.

Carpals (Fig. 11)—The left carpals are well-preserved and in loose articulation while the right carpals are preserved disarticulated with each other. Hence, our description is based on the left carpals. The carpals are composed of the ulnare, radiale, intermedium, and one distal carpal, and all are visible in dorsal view. The ulnare is a cranially convex, pyramidally-shaped block that articulates proximally with the ulna and distally with metacarpal V. The intermedium is located medial to the ulnare and is a square block with a proximally convex margin that articulates proximally with the ulna and distally with metacarpal II. Compared to the ulnare, the radiale is similar in shape but smaller in size. The radiale articulates proximally with the radius and distally both with metacarpal I and II. The distal carpal is pentagonal in outline, dorsally convex, and articulates with the third metacarpal ventrally.

Metacarpal (Fig. 11)—All five of the left metacarpals are preserved in relative articulation while the right metacarpals are preserved but disarticulated with each other. The description is based on the left metacarpals that are visible in ventral view. Proximal ends of the metacarpals are expanded and flattened to meet with the carpals. Metacarpal 1 is mediolaterally compressed whereas metacarpals 2–5 are well-preserved. All metacarpals are observable in ventral view. The second metacarpal is the longest, followed by the third and the fourth. The fifth is the shortest. The proximal ends of the metacarpals are blunt but expanded. The shafts are constricted with the shaft of metacarpal 3 being constricted most abruptly. The distal ends of the metacarpals are moderately expanded with ginglymoid articulations. On the distal ends, the lateral condyles are larger than the medial condyles, making the metacarpals medially oriented. A shallow, oval fossa exists between the distal condyles on the ventral surfaces of each distal metacarpal.

Carpal phalanges (Fig. 11, Table 4)—The left carpal phalanges are preserved in articulation while the right metacarpals are preserved in disarticulation with each other. Hence, the description is based on the left carpal phalanges. The carpal phalanges are observable in lateral view. Pulaosaurus’s phalangeal formula is 2-3-4-3-2. The proximal-most phalanx of the digit 2 is the longest. Except for the unguals, the phalanges have ginglymoid articulations with each other. The unguals of manual digits 2 and 3 are claw-like, the ungual of manual digit 1 is subconical, and the unguals of manual digits 4 and 5 are blunt.

Ilium (Figs. 12A and 12B, Table 5)—Only the right ilium is visible in IVPP V30936, which is preserved upside down. The ilium is composed of an elongate and tapering preacetabular process, the main body, and a dorsoventrally deep, rostrocaudally short postacetabular process. The ischial peduncle is also compressed and damaged, so the exact shape of the ischial peduncle is unknown. Similar to Dryosaurus and other neornithischians (Galton, 1981; Han et al., 2018), the preacetabular process is elongate, dorsoventrally narrow, and tapering. The preacetabular process is longer than the postacetabular process. The shape of the dorsal border is uncertain as it may be deformed. The acetabulum is rostrocaudally long and dorsoventrally narrow. On the rostroodorsal corner of the acetabulum, there is a supraacetabular crest that extends moderately laterally, which is a synapomorphy only seen in early-diverging ornithischians including Agilisaurus (Peng, 1992) and Eocursor (Butler, 2010) but absent in Heterodontosauridae (Butler, 2010). The pubic peduncle is rostroventrally oriented and tapers into a stout and subrectangular process in lateral view. The ischial peduncle is also anteroventrally oriented with a sub-oval articulate surface. A sub-oval and shallow brevis fossa facing ventrolaterally is present on the ventral margin of the postacetabular process, the width of which is about eight mm.

Figure 12 The pelvic girdle in lateral view.

(A) The photograph of the left ilium in lateral view. (B) The line drawing of the left ilium in lateral view. (C) The photograph of the left pubis and the left ischium in lateral view, the missing part supplemented by the line drawing. (D) The CT scaning image of the left pubis and the left ischium. Abbreviations: bf-brevis fossa, bsopp-the bony sheet on the proximal pubis, ip-the ischial peduncle, isc, ischium; opi, the obturator process of the ischium; pop, the postacetabular process; prep, the preacetabular process; pup, the pubic peduncle; sc, the supra-acetabular crest. Photo credit: Hailong Zhang.

Ischium (Figs. 12C and 12D)—The left ischium is preserved with its proximal end overlapped by a rib and the left femur distal end overlapped by the left tibia. The ischium is a rod-like element that contacts the pubis rostroventrally. The proximal end is divided into the pubic peduncle and the iliac peduncle. In lateral view, the pubic peduncle is anterodorsally oriented while the iliac peduncle is dorsally oriented. The former is rostrocaudally broader and longer than the latter. The concavity between both peduncles is dorsoventrally shallow and rostrocaudally broad. The obturator process is small and subrectangular. Similar to the ischium of Gilmoreosaurus mongoliensis (AMNH FARB 30739) (Prieto-Márquez & Norell, 2010), the obturator process is adjacent to the pubic peduncle and forms the obturator foramen with the pubis. The ischial shaft is in caudoventral orientation with a weakly expanded distal end.

Table 5 Measurements of Pulaosaurus qinglong specimen V30936 pelvic girdle.

Total length of the ilium	69.84 mm	
Depth of the blade above the acetabulum	18.88 mm	
The width between the rostral margin of the pubic peduncle and the caudal margin of the ischial peduncle at the base	23.62 mm	
Preacetabular process length	28.08 mm	
Postacetabular process length	21.90 mm	
Postacetabular process depth	21.16 mm	

Pubis (Figs. 12C and 12D)—The left pubis is preserved with its proximal end overlapped by the right femur. It is a slender, mediolaterally thin, rod-like bone in caudoventral orientation. The length of the pubis shaft is subequal to the left ischium. It contacts the left ischium on the dorsal border of the shaft. Compared to the ischium, the pubic body is smaller. Due to overlapping, the prepubic process is not visible. A small bony sheet on the proximal pubic shaft indicates that the obturator opening in the pubis is a notch-shape and forms the obturator foramen with the ischium, as observed in Haya (Barta & Norell, 2021), Jeholosaurus (Han et al., 2012), and Thescelosaurus (Brown, Boyd & Russell, 2011).

Femur (Fig. 13, Table 6)—Both femora are preserved in medial view, so the greater and lesser trochanters are not visible. The distal end of the left femur is broken while the right femur is compressed with its fourth trochanter broken. The femur is robust with an elliptical cross-section and is longer than the humerus. Cranially, the femoral head is perpendicular to the shaft which is bowed rostrally. The neck between the head and the shaft is not visible. The fourth trochanter is pendant-shaped and is located entirely on proximal half of the shaft. Although the distal end is broken, eroded, or compressed, the medial condyle of the distal femur is larger than the lateral condyle based on the morphology of the intercondylar groove. However, due to incomplete or deformed preservation of the distal end, the exact shape of the intercondylar fossa is unknown.

Figure 13 The femora of Pulaosaurus qinglong (IVPP V30936).

(A) The left femur in medial view. (B) The right femur in medial view. Photo credit: Hailong Zhang.

Table 6 Measurements of Pulaosaurus qinglong V30936 femora, tibiae and fibula.

Left femur length in medial view	88.52 mm	
Proximal end width of left femur in medial view	14.04 mm	
Distal end width of left femur in medial view	14.68 mm	
Shaft minimal width of left femur in medial view	11.80 mm	
Right femur length in medial view	80.94 mm	
Proximal end width of right femur in medial view	?	
Distal end width of right femur in medial view	16.24 mm	
Shaft minimal width of right femur in medial view	?	
Left tibia length in anterior view	97.84 mm	
Proximal end width of left tibia in anterior view	20.02 mm	
Distal end width of left tibia in anterior view	?	
Shaft minimal width of left tibia in anterior view	10.94 mm	
Right tibia length in medial view	98.76 mm	
Proximal end width of right tibia in medial view	20.44 mm	
Distal end width of right tibia in medial view	16.98 mm	
Shaft minimal width of right tibia in medial view	9 mm	
Left fibula length in anterior view	94.90 mm	
Proximal end width of left fibula in anterior view	2.68 mm	
Distal end width of left fibula in anterior view	7.3 mm	
Shaft minimal width of left fibula in anterior view	1 mm	

Tibia (Fig. 14, Table 6)—Both tibiae are preserved. The left tibia is observable in lateral view while the right tibia is observable in medial view with both being elongated, rod-like bones. The tibia is longer than the femur with the ratio between the left tibia and the left femur being 1.10 while the ratio between the right tibia and femur is 1.22. Both ratios are similar to most early-diverging neornithischians (Barta & Norell, 2021) and the ratio would become smaller in the more mature individuals (Barta & Norell, 2021). Due to displacements of the left fibula and tarsals, the left tibia distal end overlapped by the fibula, thus obscuring the exact contacting relationship between the left tibia and the left fibula. In lateral view, the proximal end of the tibia extends rostrally. The cnemial crest is poorly developed and forms a short, anterolaterally extending ridge that is separated from the fibular condyle by the deep incisura tibialis. In lateral view, the fibular condyle is prominent and broader than the cnemial crest and extends lateral and proximally to form a proximodistally extending fibular crest. A tiny accessory condyle lies medial to the fibular condyle. The shaft of the tibia is elongated and the cross-section is elliptical in shape. The shaft expands moderately laterally to form a laterally convex ridge, the fibula eminence, to contact the fibula. In medial view, the distal end of the tibia expands anteroposteriorly and in anterior view it expands mediolaterally to form the internal malleolus and the external malleolus. A shallow groove on the lateral side tibia separates the internal malleolus and the external malleolus. The external malleolus is obscured by the distal end of the left fibula while the medial malleolus on the distal end of the right tibia extends medially and anteriorly.

Figure 14 The tibiae and the fibulae.

(A) The left tibia and the left fibula in view. (B) The right tibia and the right fibula in medial view. Photo credit: Hailong Zhang.

Fibula (Fig. 14)—Both fibulae are preserved, but only the left one is observable in lateral view. The shaft of the observable fibula bows laterally. It lies lateral to the left tibia and is unfused with the left tibia. However, due to its displacement, the exact contact relationship between the fibula and the tibia is unknown. Compared to the left tibia, the length of the fibula is subequal to the length of the tibia while the fibula is much more gracile. In the lateral view, the proximal end of the fibula is narrow and extends anteromedially. From the midpoint to the distal third of the fibula, the shaft narrows. The distal end of the fibula gradually expands mediolaterally.

Astragalus and calcaneum (Fig. 15, Table 7)—The astragalus and calcaneum of the right hindlimb are both preserved in anterior view. In anteroventral view, the right astragalus and calcaneum contact the distal end of the right tibia proximally and are tightly appressed without co-ossifying. The left astragalus is displaced, but it is visible in dorsal view. The astragalus and calcaneum are not fused to the distal end of tibia, similar to the condition seen in Hypsilophodon (Huxley, 1870), Orodromeus (Scheetz, 1999), Haya (Barta & Norell, 2021), Changchunsaurus (Butler et al., 2011), Jeholosaurus (Han et al., 2012), and Oryctodromeus (Krumenacker, 2017). The length of right astragalus ventral margin is about three times the length of right calcaneum ventral margin.

Figure 15 The right and left astragali, the right and left pedes and the right calcaneum.

(A) The right astragalus and calcaneum in anterior view, right metatarsals in dorsal view and the phalanges in lateroventral view. (B) The left astragalus in dorsal view, the right distal tarsals and metatarsals in ventral view and the right phalanges in lateral view. (C) The left calcaneum in anterior view. Abbreviations: ast-astragalus, cal-calcaneum, dt-distal tarsal, mt-metatarsal, I-1-phalanx I-1, I-2-phalanx I-2, II-1-phalanx II-1, II-2-phalanx II-2, II-3-phalanx II-3, III-1-phalanx III-1, III-2-phalanx III-2, III-3-phalanx III-3, III-4-phalanx III-4, IV-1-phalanx IV-1, IV-2-phalanx IV-2, IV-3-phalanx IV-3, IV-4-phalanx IV-4, IV-5-phalanx IV-5. Photo credit: Hailong Zhang.

The astragalus is sub-rectangular in dorsal view and subtriangular in anterior view. Only the dorsally extending anterior face and the concave dorsal astragalus surfaces are visible. The anterior face of the astragalus forms the ascending process, which is represented as a subtriangular flange that extends dorsolaterally. This character is also seen in Gilmoreosaurus mongoliensis (Ruiz-Omeñaca et al., 2012). The lateral margin of the astragalus is moderately convex to contact the calcaneum while the medial margin is concave and longer. A small oval articulating surface for the distal end of the fibula is located on the lateral margin, at the base of the ascending process. The proximal surface is dorsoventrally concave and forms an elliptical fossa that articulates with the tibia.

Table 7 Measurements of the right astragalus, the right calcaneum and the left astragalus.

The length of right astragalus ventral margin	18.42 mm	
The length of right calcaneum ventral margin	6.40 mm	
The height of the ascending process on the anterior side of the right astragalus	4.58 mm	
The length of left astragalus ventral side’s anterior margin	11.10 mm	
The length of left astragalus ventral side’s posterior margin	11.83 mm	
The height of the ascending process on the anterior side of the left astragalus	4.10 mm	
The anteroposterior width of the left atragalus	9.16 mm	

The calcaneum is square and is only visible in anterior view. Due to poor preservation, little information about the calcaneum is available.

Distal tarsals (Fig. 15)—There are three distal tarsals preserved, which is the same number observed in Heterodontosaurus (Han et al., 2012). This is possibly an ornithischian plesiomorphy (Norman et al., 2011; Peng, 1992). Both left and right distal tarsal 1 are preserved. The left distal tarsals 2 and 3 are preserved. Distal tarsal 1 is situated above metatarsal 1 and 2, distal tarsal 2 is situated above metatarsal 3 and 4, distal tarsal 3 is situated above metatarsal 4, and metatarsal 5 is dorsolaterally adjacent to distal tarsal 3. Distal tarsal 1 is a wedge-shaped element with a with a shallow fossa on its posterior surface. In posterodorsal view, distal tarsal 1 is an L-shaped element with a concave posterior surface. The dorsoventral height of distal tarsal 1 narrows from medial to lateral. In the posterior view, distal tarsal 2 is a block-like element, the mediolateral width of which is much longer than its dorsoventral length. In the posterior view, distal tarsal 3 is a drop-shape element, the mediolateral width of which narrows from dorsal to ventral. A small foramen pierces the posterior surface of distal tarsal 3, a feature also observed in the posterior surface of distal tarsal 3 in Jeholosaurus (Han et al., 2012).

Metatarsals (Fig. 15, Table 8)—The metatarsals are preserved as mostly complete elements. Right metatarsals I, II, III, and IV are visible in dorsal view while left metatarsals II, III, IV, and V are visible in ventral view. The proximal end of right metatarsal I is obscured by right metatarsal II. The midpoint and distal end of right metatarsal III are damaged and the distal half of right metatarsal IV is not preserved. Metatarsal V is broken. Only the proximal end is preserved and it is dorsolaterally appressed to distal tarsal 3. Metatarsals are almost in the same plane and appressed to each other throughout most of their lengths. As in most early-diverging neornithischians, metatarsal III is the longest and the stoutest with its length being about twice the length of metatarsal I. Metatarsal I is splint-like with its proximal end mediolaterally compressed and its shaft gradually widening proximodistally. In dorsal view, metatarsal II, III and IV are elongated, rod-like elements with the shafts proximodistally narrow and their distal ends moderately expanded. Shallow grooves, ginglymoid distal articular surfaces, and collateral ligament pits are present on the dorsal sides of distal ends. In the ventral view, medial condyles on the distal ends of metatarsal II, III and IV are larger than the lateral condyles.

Table 8 Measurements of Pulaosaurus qinglong V30936 metatarsals and pedal digits.

Left/Right	Digit	Perspective	Length/mm	Proximal end width/mm	Distal end
width/mm	
Right	Metatarsal I	anterior	27.68	?	5.64	
Right	I-1	anterior	16.23	5.36	3.61	
Right	I-2	anterior	10.06	4.66	/	
Right	Metatarsal II	anterior	47.10	6.58	4.86	
Right	II-1	anterior	16.82	7.44	4.96	
Right	II-2	lateral	14.5	6.46	4.7	
Right	II-3	lateral	12.9	5.28	/	
Right	Metatarsal III	anterior	53.86	5.6	11.6	
Right	III-1	posterior	15.68	11.42	8.04	
Right	III-2	posterior	12.28	8	6.78	
Right	III-3	posterior	12.34	8.12	7.22	
Right	III-4	posterior	15.66	6.22	/	
Right	Metatarsal IV	anterior	?	6.42	?	
Right	IV-1	anterior	?	?	5.48	
Right	IV-2	lateral	8.86	6.44	4.22	
Right	IV-3	lateral	7.96	5.6	5.02	
Right	IV-4	lateral	6.6	5.16	3.74	
Right	IV-5	lateral	11	4.32	/	
Left	Metatarsal II	posterior	47.28	8.32	7.42	
Left	II-1	posterior	16.46	5.72	5.32	
Left	II-2	anterior	?	8.2	?	
Left	II-3	anterior	12.8	3.22	/	
Left	Metatarsal III	posterior	54.36	6.92	9.52	
Left	III-1	anterior	18.06	8.32	7.62	
Left	III-2	anterior	14.71	8.28	7.36	
Left	III-3	lateral	12.72	6.5	5.5	
Left	III-4	lateral	16.36	5.41	/	
Left	Metatarsal IV	posterior	66.04	8.52	5.56	
Left	IV-1	anterior	11.53	7.52	6.08	
Left	IV-2	anterior	8.9	5.53	5.38	
Left	IV-3	anterior	8.6	4.92	4.26	
Left	IV-4	lateral	7.32	4.48	4.28	
Left	IV-5	lateral	12.04	5.90	/	

Pedal phalanges (Fig. 15)—Right pedal phalanx I, II, III, IV, and left phalanx II, III, IV are preserved. Due to poor preservation, it is uncertain whether phalanx V exists in this taxon. Based on the preserved digits, the phalange formula of Pulaosaurus is 2-3-4-5-?, which is similar to most early-diverging ornithopods and early-diverging ceratopsians (Norman et al., 2004; You & Dodson, 2004). Most of the left digits are observable in the dorsal view except for the ungual digits of phalanx II and IV which are visible in the lateral view. Most of the right digits are observable in the ventral view except for the ungual digits of phalanx II, III, and IV which are visible in the lateral view, and phalanx I which is visible in the dorsal view. Right phalanx I and IV, and left phalanx III are disarticulated from their metatarsals. Except for ungual digits, the proximal and distal ends are expanded while the shafts are constricted. In dorsal view, except for III-1 and III-2, all of the left digits have dorsal lips on proximal ends that articulate with the extensor grooves of the preceding phalange distal ends. Deep oval collateral ligament pits on distal ends of the dorsal surface of III-1 and III-2.On the ventral sides of the right phalanges, lateral and medial condyles on distal ends are subequal. The intercondylar grooves are deep. All ungual digits preserved are claw-like. The ungual of phalanx III is longest, followed by phalanx II, IV and I (Table 8). The ungual of phalanx II has the greatest curvature among the pedal unguals while others are subequal.

Gut contents (Fig. 16)—On the posteroventral corner of the specimen’s thoracic cavity, variegated impressions and flat pebbles can be found. The shape of these impressions is not the same as each other and the diameter of these impressions ranges from three mm to eight mm, obviously larger than the diameter of the surrounding matrix. The exact number of these impressions is unknown as it is hard to distinguish them from the surrounding matrix and their outlines are ambiguous. Different morphological characters and scattered distribution among these impressions make it impossible to be ovarian follicles as ovarian follicles that are preserved in fossil enantiornithine specimens are circular and uniform in size (Bailleul et al., 2020; O’Connor et al., 2014; Wang et al., 2016). These oval or elliptical marks and flat peddles are similar to impressions and cavities preserved in Minmi which are considered to be left by the displaced plant seeds (Molnar & Clifford, 2000).

Figure 16 The gut contents preserved in IVPP V30936.

P-Pebbles; I-Impressions of possible plant seeds. Photo credit: Hailong Zhang.

Therefore, it is possible that the gut contents, which are likely plant seeds, are preserved in this specimen. However, the actual identity of this thoracic anomaly needs further study that is beyond the scope of this project.

Discussion

Phylogenetic analysis

The first analysis produced 26,901 most parsimonious trees of 1,225 steps, a consistency index (CI) of 0.36, and a retention index (RI) of 0.71. The resolution of the strict consensus tree (Fig. 17) is lower than the strict consensus tree by Han et al. (2018). Most clades supported by the previous strict consensus tree by Han et al. (2018), except for Heterodontosauria, Pachycephalosauria, Iguanodontia, Neoceratopsia, Thyreophora, are not supported by this strict consensus tree. Most early-diverging ornithischian taxa and early-diverging neornithischian taxa form a polytomy which makes it difficult to recover the phylogenetic relationships between early-diverging ornithischian and early-diverging neornithischian taxa, especially the phylogenetic position of Pulaosaurus. This is possibly because of the incompleteness of morphological characters found in unstable taxa, as the results and resolution of different phylogenetic analyses also vary with the sampling taxa, the sampling characters and the completeness of specimens (Brown et al., 2022).

Figure 17 The strict consensus tree from 26,901 most parsimonious trees including 74 taxa and 380 characters generated by the analysis.

Nodes: 1-Ornithischia, 2-Heterodontosauria, 3-Thyreophora, 4-Pachycephalosauria, 5-Iguanodontia, 6-Neoceratopsia.

The reduced consensus tree (Fig. 18) is from 572 of the most parsimonious trees of 1,210 steps, with a consistency index (CI) of 0.37 and a retention index (RI) of 0.71. The resolution of the reduced consensus tree is much higher compared to the strict consensus tree. It supports most clades established by previous analyses (Han et al., 2018). The reduced consensus tree recovers Pulaosaurus as one of the most early-diverging taxa of neornithischians together with Agilisaurus. This result agrees on the topology of previous analyses (Boyd, 2015; Han et al., 2018; Dieudonné et al., 2020; Butler, Upchurch & Norman, 2008).

Figure 18 The reduced consensus tree from 572 most parsimonious trees including 70 taxa and 380 characters generated by the analysis.

Nodes: 1-Ornithischia, 2-Heterodontosauridae, 3-Genasauria, 4-Thyreophora, 5-Neornithischia, 6-Cerapoda, 7- Pachycephalosauria, 8-Ornithopoda, 9-Iguanodontia, 10-Ceratopsia, 11-Chaoyangsauridae, 12-Neoceratopsia.

The monophyly of Neornithischia is supported by the combination of following synapomorphies: buccal emargination on the maxilla; both of the quadrate condyles are subequal in size; the frontal does not participate in the supratemporal fenestra; a longitudinal ridge is present along the basioccipital; a well-developed coronoid process is present on the mandible; the external mandible fenestra is absent; the prepubic process is rod-like or dorsoventrally compressed. The combination of following synapomorphies support the monophyly of Cerapoda: enamel distribution on cheek teeth is asymmetric; the fossa trochanteris is modified into a distinct constriction that separates the head and the greater trochanter of the femur; the anterior trochanter of the femur is closely appressed to the greater trochanter. Based on these phylogenetic analysis results, as the anterior side of Pulaosaurus femur and the sacrum are not visible in this specimen, it is possible that Pulaosaurus could be a member of Cerapoda.

The second phylogenetic analysis with weight equal based on Fonseca et al. (2024) dataset produces 10,000 most parsimonious trees of 7,085 steps, a consistency index (CI) of 0.16, and a retention index (RI) of 0.62. The analysis recovers Pulaosaurus as the earliest-diverging neornithischian (Fig. 19), a result similar to that produced by the first analysis. The analysis with implied weight produces 99,300 most parsimonious trees of a score, 285.09. Its CI is 0.16 while the RI is 0.62. This analysis recovers Pulaosaurus as one of the earliest-diverging neornithischians, a sister taxon to Sanxiasaurus (Fig. 20).

Figure 19 The strict consensus tree generated by analysis 2 based on the character matrix dataset provided by Fonseca et al. (2024).

Figure 20 The strict consensus tree generated by the analysis based on the dataset of Fonseca et al. (2024) with implied weight K=12.

Compared to later-diverging neornithischians, many plesiomorphies of Neornithischia are maintained in Pulaosaurus: there are five premaxillary teeth; the first maxillary tooth is close to the posterior margin of the premaxilla; the maxillary teeth are subtriangular without ridges on the labial surfaces; the forelimbs are relatively shorter compared to the hindlimbs; there are five digits on the manus; the manual ungual digits are claw-like in shape; a supra-acetabular crest is located on the ilium; there are three unfused distal tarsals. The above listed plesiomorphies suggest Pulaosaurus is one of the earliest-diverging neornithischians. However, there are also derived synapomorphies present in Pulaosaurus. For example, the posterolateral process of the Pulaosaurus premaxilla contacts the lacrimal, which is a synapomorphy of iguanodontian taxa such as Dryosaurus, Iguanodon, and Ouranosaurus (Norman et al., 2004); the frontals of Pulaosaurus are elongated and narrow which is more similar to early-diverging Ornithopoda taxa (Norman et al., 2004); the dorsal surface of the squamosal is flat and expands laterally, which is a synapomorphy of Pachycephalosauria and certain early-diverging ceratopsians such as Yinlong and Huayangceratops (You & Dodson, 2004). These features suggest that mosaic acquisition of traits has occurred in the course of neornithischian evolution and some synapomorphies of later-diverging neornithischian taxa have appeared early in basal taxa. However, it should be noted that the results described herein are unstable due to the incompleteness of specimens and the lack of certain morphological characteristics. To solve the problems about early-diverging neornithischian phylogeny and recover the actual systematical position of Pulaosaurus, more complete specimens are required in future research.

New information of distribution of early-diverging neornithischian taxa

Taxa of early-diverging Neornithischia in China are mainly found in Middle Jurassic strata throughout southwestern China: Agilisaurus (Peng, 1992), Hexinlusaurus (Barrett, Butler & Knoll, 2005), Yandusaurus (He & Cai, 1984) are found in Lower Shaximiao Formation of Province Sichuan and Sanxiasaurus (Li et al., 2019) are found in Xintiangou Formation of Chongqing. Agilisaurus and Hexinlusaurus are considered as the most early-diverging neornithischian taxa by most analyses (Boyd, 2015; Dieudonné et al., 2020; Butler, Upchurch & Norman, 2008) while Sanxiasaurus is the earliest record of Neornithischia in Asia (Li et al., 2019). The fossil record of Neornithischia in northern China is limited but includes: Jeholosaurus (Han et al., 2012) and Changmiania (Yang et al., 2020) found in Province Liaoning, Changchunsaurus (Jin et al., 2010) found in Province Jilin, all of which are from Lower Cretaceous strata (Yang et al., 2020; Jin et al., 2010; Han et al., 2012). There is a temporal and geographical gap between early-diverging taxa found in the Middle Jurassic strata of southwestern China and late-diverging taxa found in the Lower Cretaceous strata of northeastern China. The missing evolutionary link of Neornithischia is an early-diverging neornithischians in the Late Jurassic strata between southwestern and northeastern China. Pulaosaurus helps to serve as this ‘missing link’ as it is found in the Late Jurassic strata found in Province Hebei, which is between southwestern and northeastern China, helping fulfill the temporal and geographical gap. Moreover, ornithischian taxa that are present in Jurassic fauna, including the Shishugou and Shaximiao faunas, are missing in the Yanliao Biota (Liu, Wu & Han, 2022), which usually play the role of small and middle-bodied herbivores in the Mesozoic ecosystems. Pulaosaurus is found in the Tiaojishan Formation of Province Hebei where it is geographically situated between southwestern China and northeastern China. The most recent dating of the bottom of the Tiaojishan Formation in the Xuanhua District—Zhuolu County area of Zhangjiakou City, produces an age of approximately 164. 3 ± 2. 6 Ma (Bai et al., 2024). Alternatively, Wu et al. (2024) proposed an age range of 153 Ma to 162 Ma based on zircon U-Pb dating of the samples from borehole Yang D1 in Western Liaoning within the Tiaojishan Formation (Wu et al., 2024). Regardless of the dating study, the age of Tiaojishan Formation is reliably recovered as being deposited between the Late Middle Jurassic to the Early Late Jurassic. As a result, the discovery of Pulaosaurus fills the temporal and geographic gap of neornithischian fossil record in China, and provides new information on the biodiversity of the Yanliao Biota. Huang (2015) proposed that there was geographical isolation between northeastern China and other regions during the Middle and Late Jurassic, which inhibited the dispersal of organisms between faunas and made the species composition of the Yanliao Biota unique from other Jurassic faunas (Liu, Wu & Han, 2022). However, the discovery of Pulaosaurus in Upper Jurassic Tiaojishan Formation also suggests that Asian neornithischian taxa originated near or within the Middle Jurassic-aged Sichuan Basin (Li et al., 2019) and spread to northern China during the Late Jurassic and the Early Cretaceous (Yang et al., 2020; Jin et al., 2010; Han et al., 2012). This indicates that the geographical isolation of the Yanliao Biota was not as large of a preventative faunal barrier between other regions as hypothesized and the biodiversity of the Yanliao Biota has been severely underestimated.

Hyolaryngeal apparatus and acoustic function of Pulaosaurus

The hyolaryngeal apparatus of Archosauria comprises the following elements: one basihyal, a pair of ceratohyals, one pair of ceratobranchials, one pair of cricoids, and one pair of arytenoids (Yoshida, Kobayashi & Norell, 2023; Friedman et al., 2018; Hill et al., 2015) (Fig. 21). Additionally, one procricoid, one pair of epibranchials, and one paraglossal are only present in Aves (Yoshida, Kobayashi & Norell, 2023; Friedman et al., 2018; Hill et al., 2015). In extant reptiles, the hyolaryngeal apparatus elements are cartilaginous with the exception of the ossified ceratobranchials (Yoshida, Kobayashi & Norell, 2023). In extant Aves, the ceratobranchials, the epibranchials, and the larynx are ossified (Yoshida, Kobayashi & Norell, 2023). The hyolaryngeal apparatus plays a significant role in acoustic function, airway protection, respiratory modification, and circulation assistance in tetrapods (Kirchner, 1993). However, few fossilized larynx elements have been found in non-avian reptile fossils when compared to their significance in Archosauria evolution and ecology. In the case of the Dinosauria, most non-avian dinosaur specimens only preserve the first pair of rod-like ceratobranchials. Other hyolaryngeal elements preserved areas following. In Carnotaurus, Microraptor, Confuciusornis, the basihyal has been recovered (Bonaparte, Novas & Coria, 1990; Yoshida, Kobayashi & Norell, 2023); the basihyal and ceratohyals are preserved in Saichania chulsanensis (Bonaparte, Novas & Coria, 1990; Yoshida, Kobayashi & Norell, 2023); a pair of ceratobranchials and the epihyal are preserved in Pinacosaurus granger (Morschhauser & Lamana, 2013); the plate-like second pair of ceratobranchials is preserved in Psittacosaurus mongoliensis (Morschhauser & Lamana, 2013); the splint-like second pair of ceratobranchials is preserved in Leptoceratops gracilis (Morschhauser & Lamana, 2013); the tetraradiate first pair of ceratobranchials and the plate-like second pair of ceratobranchials are preserved in Protoceratops andrewsi (Morschhauser & Lamana, 2013); and the sigmoid ceratohyal and the basihyal are preserved in Triceratops horridus (Morschhauser & Lamana, 2013). Of note, Pinacosaurus granger was the first dinosaur whose larynx element has been reported (Yoshida, Kobayashi & Norell, 2023). Here, Pulaosaurus is the second reported non-avian dinosaur specimen with a preserved larynx apparatus since Pinacosaurus granger, which demonstrates that an ossified hyolaryngeal apparatus has existed more taxonomically broadly among non-avian dinosaurs rather than just in ankylosaurids. Pulaosaurus also possesses a pair of rod-like ceratobranchials, which is also preserved in Jeholosaurus and many other dinosaur species (Friedman et al., 2018).

Figure 21 The hyoaryngeal apparatuses in different taxa of Archosauria.

(A) The line drawing of Pinacosaurus grangeri (IGM100/3186) hyolaryngeal apparatus based on the 3D reconstruction. (B) The line drawing of Pulaosaurus qinglong (IVPP V30936) hyolaryngeal apparatus. (C) The line drawing of Nothoprocta sp. (AMNH6502) hyolaryngeal apparatus. (D) The line drawing of Tomistoma schlegelii (AMNH R113078) hyolaryngeal apparatus. All the specimens involved except for IVPP V30936 are from the supplementary materials provided by Yoshida, Kobayashi & Norell (2023). Abbreviations: AP-Arytenoid Process, Ary-Arytenoid, Bh-Basihyal, Cera-Ceratobranchial, Cr-Cricoid, Dlw-Dorsolateral wing, Dmw-Dorsomedial wing, Epi-Epibranchial, lw-the laterodorsal wing, mw-the mediodorsal wing. Illustration credit: Junki Yoshida, Ph.D.

The arytenoids of Pulaosaurus are elongated with arytenoid processes. The length of the arytenoids is about 80% of the dentary length (Table 1). This structure is similar to the arytenoids preserved Pinacosaurus, but the arytenoid processes of Pulaosaurus are less prominent. In the case of acoustic function, the larynx functions differently between extant Aves and non-avian reptiles (Yoshida, Kobayashi & Norell, 2023). In extant non-avian reptiles—such as turtles and crocodiles—the larynx functions as the vocal source (Yoshida, Kobayashi & Norell, 2023; Sacchi et al., 2004; Riede et al., 2015). During phonation, the glottis is almost closed by the larynx and its surrounding muscles and ligaments and then air pressure forces the glottis to open, making vocal folds to vibrate and phonate, thus producing sounds (Yoshida, Kobayashi & Norell, 2023; Sacchi et al., 2004; Riede et al., 2015). In extant Aves, the vocal source is the syrinx, which is located at the inferior end of the trachea (Kingsley et al., 2018; Yoshida, Kobayashi & Norell, 2023; Sober et al., 2019), which has also been found in the Mesozoic bird Vegavis (Clarke et al., 2016). The larynx serves as part of the vocal resonator tract to improve vocal efficiency and sounds are emitted through it, which requires control over the glottal opening (Kingsley et al., 2018; Yoshida, Kobayashi & Norell, 2023; Sober et al., 2019). A longer arytenoid provides more attachment area for the dilator muscle, thus making its lever arm longer, which assists the arytenoid with the opening of the glottis (Yoshida, Kobayashi & Norell, 2023). The arytenoids, with their prominent arytenoid processes and firm cricoid-arytenoid joints, allow for the horizontal rotation of the arytenoid to open and close the glottis (Yoshida, Kobayashi & Norell, 2023). Such structures allow extant birds to communicate with more complicated sounds in broader vocal ranges and with greater efficiency (Yoshida, Kobayashi & Norell, 2023; Sober et al., 2019). Yoshida, Kobayashi & Norell (2023) proposed that the arytenoid length is positively correlated to the mandible width and there is a distinction in the relative arytenoid size compared to the mandible between the group of the larynx vocal source and the group of the larynx vocal modifier (Yoshida, Kobayashi & Norell, 2023). Due to the compression of the Pulaosaurus mandible, the exact width of the mandible is unknown, so acoustic calculations of Pulaosaurus cannot be made. However, based on the cranial morphology, the Pulaosaurus mandible width is less than eight cm, which is the length of its skull. Therefore, the mandible width is shorter than the mandible width of Pinacosaurus, which is 10 cm (Hill et al., 2015). Pulaosaurus arytenoids are subequal to those of Pinacosaurus in length (Yoshida, Kobayashi & Norell, 2023). Therefore, the relative arytenoid length of Pulaosaurus is larger than Pinacosaurus. Pulaosaurus is likely to possess a non-laryngeal vocal source similar to Pinacosaurus although the acoustic function is more primitive as its arytenoid processes are less prominent. The larynx of Pulaosaurus possibly functions to modulate and enhance sounds, thus allowing Pulaosaurus to communicate with more complicated sounds, similar to extant birds (Yoshida, Kobayashi & Norell, 2023). This suggests that a non-laryngeal vocal source was present among the Dinosauria during at least the Late Jurassic, regardless of whether non-laryngeal vocalization is a plesiomorphy of Dinosauria or it is convergent in different dinosaur taxa (Yoshida, Kobayashi & Norell, 2023). As the fossilized syrinx could be found in the Mesozoic bird Vegavis (Clarke et al., 2016), it is possible that a fossilized syrinx could be found in non-avian dinosaur specimens in the future.

Additionally, the preservation of ossified arytenoid in Pulaosaurus strongly suggests that ossification of the laryngeal apparatus has occurred not only in Ankylosauria and Aves (Yoshida, Kobayashi & Norell, 2023) but also in Neornithischia (Fig. 22). This indicates that ossified laryngeal apparatuses should have been phylogenetically widespread among non-avian dinosaurs. However, except for Pulaosaurus and Pinacosaurus, there are no other reports of ossified laryngeal apparatus preserved in non-avian dinosaur fossils. There are two possible explanations for this paucity of laryngeal anatomy within the non-avian dinosaur fossil record. Firstly, laryngeal elements are gracile elements that rarely preserve or are taphonomically destroyed prior to discovery and description. Secondly, it is possible that other ossified laryngeal elements have been preserved, but have been misidentified (Yoshida, Kobayashi & Norell, 2023). For example, the cricoids and arytenoids of Pinacosaurus were originally incorrectly identified as the paraglossals and the first pair of ceratobranchials (Hill et al., 2015). As described above, there have been many reports that there are two pairs of ceratobranchials in many dinosaur taxa which are defined as plate-like, splint-like, or tetraradiate (Morschhauser & Lamana, 2013). However, the second pair of ceratobranchials are lost in extant archosaurs. It is possible that hyolaryngeal elements preserved in many non-avian dinosaur specimens that are currently identified as ceratobranchials are, in fact, ossified laryngeal elements. Reanalysis of vocal anatomy within non-avian dinosaurs needs to be carried out to assess the accuracy of identification among curated specimens.

Figure 22 Evolution of hyolaryngeal elements in Archosauria.

The figure is adapted from illustration by Tatsuya Shinmura (Yoshida, Kobayashi & Norell, 2023). Yellow-arytenoid, green-first pair of ceratobranchials, black-second pair of ceratobranchials, red-cricoid, white-basihyal, blue-procricoid, orange-paraglossal, grey-epibranchials. Numbers represent the ancestral state of characteristics: 1-Laryngeal vocal source, 2-Loss of second ceratobranchials, 3-Procrocoid, 4-Arytenoid process, 5- Arytenoid process, 6-Ossified larynx, 7-Immobile lungs, 8-Procrocoid, 9-Paraglossal. Illustration credit: Junki Yoshida.

Rod-like ceratobranchials are observed in Pulaosaurus. The length of Pulaosaurus ceratobranchials is less than 50% of the dentary length and their relative length is subequal to those of Jeholosaurus (Barrett & Han, 2009) but shorter than those of Paraves and quadrupedal ornithischians (Friedman et al., 2018). Elaborate ossified hyoid elements are typically observed in Aves, pterosaurs, and quadrupedal ornithischians such as ankylosaurids and hadrosauroids, which increases the mobility of the tongue and makes up for the diminished utility of forelimbs (Friedman et al., 2018). In pterosaurs, the ceratobranchials are elongated and fused (Friedman et al., 2018). In Aves, epibranchials arise to increase the overall length of the hyoid element. In quadrupedal ornithischians, the frequency of ossification of hyoid elements increases (Friedman et al., 2018). The elongation of the ceratobranchials supports the mobility of the avian tongue (Friedman et al., 2018), which is closely associated with the feeding and ecological radiation of Aves.

The short relative length of Pulaosaurus’ ceratobranchials suggests that the tongue mobility of Pulaosaurus would have been limited. This may have been because Pulaosaurus was an obligate biped with its forelimbs used for food acquisition and processing. The limited tongue mobility and primitive tooth morphology of Pulaosaurus also indicate that the food intraoral processing of Pulaosaurus is less prominent than later-diverging ornithischian taxa (Friedman et al., 2018) and it could only feed on softer food.

Conclusions

Pulaosaurus qinglong gen. et sp. nov. is an early-diverging neornithischian species found in the Upper Jurassic Tiaojishan Formation of Province Hebei, China. A phylogenetic analysis places Pulaosaurus at the base of Neornithischia close to Agilisaurus, which is the earliest-diverging neornithischian. Pulaosaurus represents the first neornithischian found in the Yanliao Biota, and helps to fill the temporal and geographical gap in the distribution of Neornithischia within China. A pair of arytenoids are preserved in the Pulaosaurus holotype and represents the second case of an ossified laryngeal apparatus among non-avian dinosaurs. The arytenoids of Pulaosaurus indicates that ossified laryngeal apparatuses were present in Neornithischia, thus suggesting that the ossified laryngeal apparatus could be widespread across Dinosauria. As the morphology of Pulaosaurus arytenoids resembles the arytenoids of extant birds, it is possible for Pulaosaurus to have an avian-like vocalization.

Supplemental Information

Supplemental Information 1 The TNT file of analysis 1

Supplemental Information 2 The TNT file of analysis 2

Supplemental Information 3 The TNT file excluding unstable taxa of analysis 1

Supplemental Information 4 The changes in morphological characters from early-diverging neornithischians to Ornithopoda

We carry out the ’common mapping of characters’ function on the the dataset ’unstable excluded’ in Supplemental Files to summarize the changes in morphological characters from early-diverging neornithischians to Ornithopoda.

We would like to thank Catherine Forster, Daniel Madzia and Filippo Bertozzo for critical and constructive reviews that drastically improved the quality of this manuscript, Hailong Zang for photographs, and Yihui Ke and Rui Pei for valuable advice on this project.

Institutional abbreviations

IVPP Institute of Vertebrate Paleontology and Paleoanthropology, Beijing, China

Additional Information and Declarations

Competing Interests

Author Contributions

Data Availability

New Species Registration

The authors declare there are no competing interests.

Yunfeng Yang performed the experiments, analyzed the data, prepared figures and/or tables, authored or reviewed drafts of the article, and approved the final draft.

James L. King conceived and designed the experiments, authored or reviewed drafts of the article, and approved the final draft.

Xing Xu conceived and designed the experiments, authored or reviewed drafts of the article, and approved the final draft.

The following information was supplied regarding data availability:

The data is available at MorphoSource: DOI: 10.17602/M2/M720739.

The TNT data is available in the Supplementary Files.

The specimen IVPP V30936 is stored in the office of Professor Xing Xu, the 9th floor of Institute of Vertebrate Paleontology and Paleoanthropology, Beijing.

The following information was supplied regarding the registration of a newly described species:

Publication LSID: urn:lsid:zoobank.org:pub:D3939AEC-9C5B-4397-9BA4-47CB3F9DFEC8.

Genus name: urn:lsid:zoobank.org:act:BF4D7285-40DD-4BBB-85F5-2F2832D30D0E.

Species name: urn:lsid:zoobank.org:act:F4F321C8-5C07-400E-A70E-FF2045FDFFE8.

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
