# Peer review of "A new neornithischian dinosaur from the Upper Jurassic Tiaojishan Formation of northern China"

_PeerJ, doi:10.7717/peerj.19664_

## Round 0.1 · original submission · Major Revisions

Although the reviewers acknowledged that your study is based on an exceptional specimen, they also identified significant issues with both the text and illustrations. In my opinion, substantial revisions are necessary in both content and presentation before it can be considered for publication in PeerJ.

·

Basic reporting

The English is very good although there are a few places where it could be improved, especially the use of the definite article "the."
The literature references and comparisons are adequate and up to date.
The structure is good and easy to follow.
See Attachment for a line by line edit.

One additional thing that would help the readers is to add a figure or two with close-up photos of regions of the skull. It is difficult to see detail on Fig. 2.

Experimental design

no comment

Validity of the findings

no comment (see attachment).

Additional comments

This is an important new taxon from a time period that is crucial to understanding the evolution of ornithischian dinosaurs.

·

Basic reporting

Thank you for the opportunity to review the manuscript by Yunfeng Yang and colleagues. The study reports a spectacular specimen that is definitely worthy of detailed description. The language of the MS is mostly clear though there are numerous issues (lacks of spaces; double dots; “it’s” instead of “it is”; “isn’t” instead of “is not”, “Neornithischian” instead of “neornithischian”, “Pinacosaurus granger” instead of “grangeri” etc.). However, these can be easily corrected during revisions or even at the proof stage and do not warrant discussion.

Relevant literature is referenced though numerous studies are incorrectly cited (e.g., “Barrett 2009” is “Barrett and Han 2009”, “Han 2017” is “Han et al. 2018”; “Berta 2021” is “Barta and Norell 2021”). But again, this is easy to correct during revisions or proofing stage. The structure is logical; however, I have problems observing some features on the figures. The preservation is very nice but the photos are not always clear enough. For example, the skull is clearly beautifully preserved but only overviews are provided. If I wanted to score the specimen for my character matrix, I would not be able to include many characters with certainty. Therefore, I would like to ask the authors to consider adding more close-ups.

Raw data (the TNT file necessary to conduct the phylogenetic analyses) are shared.

What appears to be a bit unusual is that the authors thank one of the co-author (Xing Xu) in the Acknowledgements.

Experimental design

I re-analyzed the matrix. I applied slightly different settings to explore the tree space more thoroughly (I increased the tree space, ran ‘New Technology’ search first, and then TBR using trees from RAM; I am happy to discuss that with the authors if they wish to replicate it). When using the equal weighting, I found 152280 MPTs (~7x as many as the authors; best score was the same: 1227). Additionally, I performed weighted parsimony analyses that are generally considered as providing more reliable results. I used K = 12 and K = 18 which should be OK considering the size of the matrix. Both these additional analyses placed the new taxon among neornithischians. Although the topology was not well resolved (which is expectable – ornithischians are always problematic), closer inspection of the MPTs showed that the preferred placement of Pulaosaurus is close to marginocephalians. With that said, I am not sure it is entirely appropriate to rely on a single run of phylogeny inference (but this essentially applies to all ornithischian papers) and a single dataset. I strongly suggest to explore the phylogeny more thoroughly, for example by including several methods or character matrices. The matrix assembled by Han et al. (2018) was focused on the base of Ceratopsia which might have an impact on the placement of Pulaosaurus. I would like to ask the authors to consider testing the placement of the new taxon by adding it to the matrix of Fonseca et al. (2024; https://doi.org/10.1080/14772019.2024.2346577). Such an approach may result in a much more robust phylogeny reconstruction and perhaps lead to more reliable results.

With respect to phylogenetic nomenclature, the authors mentioned that they followed Madzia et al. but in the Systematic paleontology they mentioned Sereno’s (1998) informal definition. I would remove that reference to eliminate possible confusion.

Validity of the findings

My biggest problem with assessing the morphology of the new taxon is that I cannot see some features discussed in the text. For example, the authors mentioned that one of the autapomorphies of Pulaosaurus is the absence of the coracoid foramen. Yet, the placement and development of the foramen differs in taxa; the foramen may be located on the suture between the coracoid and the scapula. When the preservation is poor, it may even seem absent. Whether it is truly absent I cannot say based on the attached figures.

The “Conclusions”, as currently written, are oversimplified and do not add much to the text. I am not sure this section (two sentences) is really needed.

Additional comments

My overall feeling is that the current version of the description would benefit from more thorough comparisons or at least high-resolution close-up figures; and also with more thorough phylogeny reconstruction. I look very much forward to seeing the new taxon published but the suggested changes, combined with the necessary corrections of the typos and cited references, would probably result in moderate to major revisions.

·

Basic reporting

Yang et al describes a new ornithischian species, named Pulaosaurus qinlong, from the Upper Jurassic Tiaojishan Formation of Qinglong, Hebei Province, Northern China.
The specimen is exquisitely preserved, and it shows a mosaic of features that can help elucidating the early relationships of basal neornithischians, one of the hottest actual topic in dinosaur paleontology. Personally speaking, the description of possible arytenoids is what makes this specimens really outstanding, not just its basal position, hence I am really in favor to see Pulaosaurus published!
The manuscript is well structured, with a proper logical flow. However, there are many problems and concerns with the style, information and methodology provided.

The English language should be improved for a better comprehension. I know that these comments can be sensitive, I myself am not a native English speaker, but too many times I had some serious difficulties understanding the structure and/or the meaning of sentences, if not paragraphs. For example, there is an over-usage of the pronom “it”. The subject for the specific description is dubious, if not lost among the many names in a sentence (see the attached pdf where I provided indications, and some times proposing modifications). I suggest to seek advice from a native speaker colleague, or someone proficient in English, to review the manuscript.
There are some problems with anatomical direction: I see that the authors decided to use anterior/posterior, but I would advise otherwise, opting for cranial/rostral and caudal. In particular, some of the bones are described as showing in anterior view when, to me, they are facing laterally. I strongly suggest to review the anatomical direction and be precise about it, since it can help the reader to navigate across the osteological descriptions.
Same goes for the tail, where proximal, middle and distal should be used instead of anterior, middle and posterior.

The literature should be addressed correctly. Many times authors and years are missing or badly reported (for example, a recurrent mistake is “Berta 2021” for Haya when it’s Barta and Norrell 2021). Furthermore, I think that the authors should view more papers about early neornithischians and basal ornithischians, especially about their phylogenetic relationships. In my opinion, the introduction is not sufficient, as a background on the early evolution of neornithischians is not present, and a presentation about the paleobiodiversity, phylogenetic relationship and biogeography of early neornithischians are missing (and just focussed on the Chinese taxa). In a few words, the reported new species is important for the early evolution of the clade, but why it is important is not specified.

Figure-wise, I have many problems with the provided pictures. The authors should add more close-up views of bones, indicating via lettering all the anatomical features the authors are highlighting in your description. In the pdf provided, I reported where the authors should do so, because many times it was impossible for me to understand what the authors were referring to. The authors have an amazing specimen, and I would love to see more about it! Given the difficulties of visualization, I strongly suggest to at least provide drawings of the figured bones to help discerning the characteristics (as you did for the skull and cervicals). For example, Fig. 13 tells me nothing. I cannot see what the authors are describing, and vectors and lettering would really help!

Another figure I think is missing is a nice osteological comparison of the arytenoids. If they are indeed arytenoids, these bones are really important in a large scale, suggesting that many ornithischian workers (me, for example) should have a re-consideration of many specimens, and it would be useful to have a detailed comparison of materials . In particular, I would like to see a figure combining the arytenoids of Pulaosaurus with those described already in other dinosaurs, and especially in birds (backed up by a more precise written comparison of the different arytenoids; what the authors reported so far is not enough, imho). Indeed, the authors provided a phylogenetic frame for it in Fig. 16, but I would like to see a more detailed visual work, with pictures (or at least drawings) of the arytenoids in the different taxa with similar/analogue/homologue features highlighted.

Experimental design

The manuscript falls within the Aims and Scope of the Journal. The main research questions, i.e. the description of a new ornithischian taxa and its phylogenetic position, are presented, but not backed up by a well-described rationale (see above for the introduction). Only the research question on the improving of the biodiversity of Yanliao Biota is supported by the presentation.

In Materials and Methods the authors provide more information about the preparation of the specimen, as well as the technical information about the CT Scanning. Furthermore, such Ct scanning... where are they? Where are they deposited? Why didn’t the authors use them in the manuscript? The authors actually showed only one (Fig.9), but there are many more figures that would require a ct visualization.

The authors describe the osteology of the specimen in details, even though many points have to be improved (see pdf).

Thanks for including the .txt file for the phylogeny. I was able to open it in TNT, but I tried to run the analysis as the authors indicate and I obtained slightly different results... Perhaps the authors should indicate the analytical steps in a more detailed way so that readers can confidently re-test the presented results.

Validity of the findings

The report of a new, basal ornithischian is a welcome addition to our knowledge of these animals, since the mysteries surrounding the base of their evolutionary tree. The suggestion that arytenoids are already well-developed in early ornithischians is a strong statement which seems to be supported by the description of the specimen. Indeed, this specimen increases the biodiversity of the Yanliao Biota, but it gives even more information on the early evolution of neornithischians.

The data provided is robust.

The conclusions are presented, but the section needs to be improved, especially with statement about the actual knowledge and implications on the phylogenetical positioning of Pulaosaurus within Neornithischia, the biogeography implications, and the arytenoid comparison with other taxa (see comments above).

Additional comments

I don’t have enough experience with phylogenetic analysis, but I have some problems with the highlighting the plesiomorphies and apomorphies in the osteological descriptions. If I understood correctly, the state of these characters is based on the authors' analysis, isn’t it? Because some seems to be taken from other papers, and references are missing. If it is based on the authors' analysis, I would avoid writing it in the description of the specimen, because that is a “Result”, while the apomorphic state is a “Discussion”. By removing them, I would add a paragraph in Discussion where the authors report apomorphies and sinapomorphies for Pulaosaurus.

There are other mistakes and problems (“palpebral” bone, paraoccipital etc), I include all of them in the supporting pdf.

In Discussion, the authors listed a loooong list of features, that adds nothing to the discussion itself. I suggest to place it in a Supplementary File (also, the numbering of the features is really confusing).

---

## Round 0.2 · Minor Revisions

Your manuscript has been re-examined by two of the original reviewers. They find that your work has improved considerably, but they still identify a number of issues that need to be addressed. I also strongly recommend that your text be thoroughly proofread to improve its clarity and readability.

**Language Note:** The Academic Editor has identified that the English language must be improved. PeerJ can provide language editing services - please contact us at [email protected] for pricing (be sure to provide your manuscript number and title). Alternatively, you should make your own arrangements to improve the language quality and provide details in your response letter. – PeerJ Staff

·

Basic reporting

I am happy to see the revised version of the manuscript and would like to thank the authors for carefully considering my suggestions. I think the MS improved considerably.

The language is mostly clear though there are still some minor issues that the authors might want to correct before publication. These are in the text (e.g., lines 1022, 1023: “there are ALSO derived synapomorphies ALSO present in Pulaosaurus”) as well as in the cited references (e.g., line 60: “Cincotta et al.” – no year provided; lines 58, 59: “Barrett and Han et al. 2005”; line 996: “Han 2017”). As I said during the first round of reviews, these are minor things and I am sure that they will be corrected before final acceptance or even during the proof stage).

All supplementary data have been provided so the analyses are replicable.

Experimental design

During the previous round I have suggested that the authors score the new taxon into the dataset of Fonseca et al. (2024). I am very happy to see that the authors did that. I have looked at the file they provided and reanalyzed the matrix. As before, I have selected different settings to explore the data more thoroughly. The authors explored it only superficially through equal weighting (EW). I analyzed it using EW as well as implied weighting (K = 12). Both analyses placed the new taxon near the base of Neornithischia though the exact placement differed a bit so a more thorough assessment would still be welcome. I would also like to note is that my analysis based on EW found many more trees than the one performed by the authors. The authors reported that “[t]he […] phylogenetic analysis based on Fonseca et al. (2024) dataset produces 200 most parsimonious trees of 7085 steps” (lines 1011, 1012). I set the max trees to 200,000 and it maxed out. So, I am sure there is >1000x more MPTs.

Validity of the findings

I expected the coracoid foramen could be found there and I am happy the authors investigated that. I have no additional comments.

Additional comments

Although the phylogenetic assessment may be more thorough and more close-ups could still be included (for example, the skull elements, as photographed and annotated, are still difficult to interpret based on the figures) I do not want to delay the publication of this manuscript for much longer. I think that, in the future, the specimen should definitely be CT-scanned and the skull and the rest of the skeleton, fully reconstructed which could provide additional insights into the anatomy of the new taxon as well as early-diverging neornithischians in general.

·

Basic reporting

The authors have presented the amended version of their manuscript about the basal ornithischian Pualosaurus qinglong, going through all the provided comments.
I applaude the big effort by the authors to fix, correct and implement suggestions. The manuscript right now is way stronger.
There are still a few minor things to be corrected, I attach the pdf with my comments.

Experimental design

There is yet one thing that needs to be amended. In the Introduction, there is a wrong assessment for the placement hypotheses for Neornithischia (from which basically all reasoning for Pulaosaurus comes from). If I have understood correctly from Brown et al. (2022), the two reported hypotheses are for the 'hypsilophodontids', not ALL Neornithischia.
I suggest the authors to read again Brown et al 2022 and fix this paragraph (from line 64 to 73). It is true that there is confusion about the phylogenetic positioning of the various early neornithischians, but be careful to not consider 'Hypsilophodontids' as synonym for Neornithischia.
I would like to have all the technical parameters of the miCT scans, like voltage for generating the pics (kV and μA), number of images and voxel size, and if you used any filters.

Validity of the findings

The description of the Pulaosaurus bones and the phylogeny are now accettable.

---

## Round 0.3 · accepted · Accept

I confirm that your paper is now accepted for publication.

·

Basic reporting

I am happy with the changes made by the authors.
Only few points (which can be fixed by the editorial group without sending the manuscript back to the authors):

Materials and methods: in the new latest corrections, "was" is preferred than "is".

Figure 12 caption: " (D) The CT scanNing image of the left pubis"

Apart from these two corrections, the paper is accepted as far as I am concerned! Congratulations!

Experimental design

n/a

Validity of the findings

n/a

Additional comments

n/a